# Improved Regularization and Robustness for Fine-tuning in Neural Networks

**Dongyue Li**
Northeastern University
li.dongyu@northeastern.edu

**Hongyang R. Zhang**
Northeastern University
ho.zhang@northeastern.edu

## Abstract

A widely used algorithm for transfer learning is fine-tuning, where a pre-trained model is fine-tuned on a target task with a small amount of labeled data. When the capacity of the pre-trained model is much larger than the size of the target data set, fine-tuning is prone to overfitting and "memorizing" the training labels. Hence, an important question is to regularize fine-tuning and ensure its robustness to noise. To address this question, we begin by analyzing the generalization properties of fine-tuning. We present a PAC-Bayes generalization bound that depends on the *distance traveled in each layer* during fine-tuning and the *noise stability* of the fine-tuned model. We empirically measure these quantities. Based on the analysis, we propose *regularized self-labeling*—the interpolation between regularization and self-labeling methods, including (i) *layer-wise regularization* to constrain the distance traveled in each layer; (ii) *self label-correction and label-reweighting* to correct mislabeled data points (that the model is confident) and reweight less confident data points. We validate our approach on an extensive collection of image and text data sets using multiple pre-trained model architectures. Our approach improves baseline methods by 1.76% (on average) for seven image classification tasks and 0.75% for a few-shot classification task. When the target data set includes noisy labels, our approach outperforms baseline methods by 3.56% on average in two noisy settings.

## 1 Introduction

Learning from limited labeled data is a fundamental problem in many real-world applications (Ratner et al., 2016, 2017). A common approach to address this problem is fine-tuning a large model that has been pre-trained on publicly available labeled data (He et al., 2019). Since fine-tuning is typically applied to a target task with limited labels, this algorithm is prone to overfitting or "memorization" issues (Tan et al., 2018). These issues worsen when the target task contains noisy labels (Zhang et al., 2016). In this paper, we analyze regularization methods for fine-tuning from both theoretical and empirical perspectives. Based on the analysis, we propose a *regularized self-labeling* approach that improves the generalization and robustness properties of fine-tuning.

Previous works (Li et al., 2018a,b) have proposed regularization methods to constrain the distance between a fine-tuned model and the pre-trained model in the Euclidean norm. Li et al. (2020) provides extensive study to show that the performance of fine-tuning and the benefit of adding regularization depend on the hyperparameter choices. Salman et al. (2020) empirically find that performing adversarial training during the pre-training phase helps learn pre-trained models that transfer better to downstream tasks. The work of Gouk et al. (2021) generalizes the above ideas to various norm choices and finds that projected gradient descent methods perform well for implementing distance-based regularization. Additionally, they derive generalization bounds for fine-tuning using Rademacher complexity. These works focus on settings where there is no label noise in the target data set. When

35th Conference on Neural Information Processing Systems (NeurIPS 2021).

label noise is present, for example, due to applying weak supervision techniques (Ratner et al., 2016), an important question is to design methods that are robust to such noise. The problem of learning from noisy labels has a rich history of study in supervised learning (Natarajan et al., 2013). In contrast, little is known in the transfer learning setting. These considerations motivate us to analyze the generalization and robustness properties of fine-tuning.

In Section 4.1, we begin by conducting a PAC-Bayesian analysis of regularized fine-tuning. This is inspired by recent works that have found PAC-Bayesian analysis correlates with empirical performance better than Rademacher complexity (Jiang et al., 2020). We identify two critical measures for analyzing the generalization performance of fine-tuning. The first measure is the $\ell_2$ norm of the distance between the pre-trained model (initialization) and the fine-tuned model. The second measure is the perturbed loss of the fine-tuned model, i.e. its loss after the model weights get perturbed by random noise. First, we observe that the fine-tuned weights remain closed to the pre-trained model. Moreover, the top layers travel much further away from the pre-trained model than the bottom layers. Second, we find that fine-tuning from a pre-trained model implies better noise stability than training from a randomly initialized model. In Section 4.2, we evaluate regularized fine-tuning for target tasks with noisy labels. We find that fine-tuning is prone to "memorizing the noisy labels", and regularization helps alleviate such memorization behavior. Moreover, we observe that the neural network has not yet overfitted to the noisy labels during the early phase of fine-tuning. Thus, its prediction could be used to relabel the noisy labels.

We propose an algorithm that incorporates layer-wise regularization and self-labeling for improved regularization and robustness based on our results. Figure 1 illustrates the two components. First, we encode layer-wise distance constraints to regularize the model weights at different levels. Compared to (vanilla) fine-tuning, our algorithm reduces the gap between the training and test accuracy, thus alleviating overfitting. Second, we add a self-labeling mechanism that corrects and reweights "noisy labels" based on the neural network's predictions. Figure 1 shows that our algorithm effectively hinders the model from learning the incorrect labels by relabeling them to correct ones.

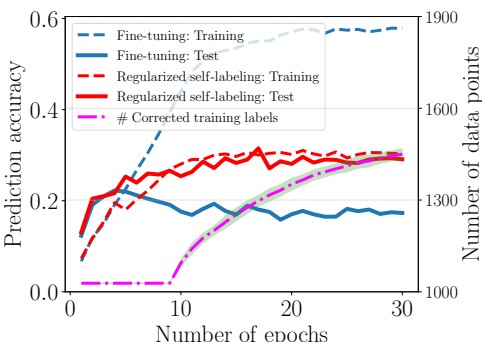

Figure 1: Red: Layer-wise regularization closes generalization gap. Magenta: Self-labeling relabels noisy data points to their correct label.

In Section 5, we evaluate our proposed algorithm for both transfer learning and few-shot classification tasks with image and text data sets. First, using ResNet-101 (He et al., 2016) as the pre-trained model, our algorithm outperforms previous fine-tuning methods on seven image classification tasks by $1.76\%$ on average and $3.56\%$ when their labels are noisy. Second, we find qualitatively similar results for applying our approach to medical image classification tasks (ChestX-ray14 (Wang et al., 2017; Rajpurkar et al., 2017)) and vision transformers (Dosovitskiy et al., 2020). Finally, we extend our approach to few-shot learning and sentence classification. For these related but different tasks and data modalities, we find an improvement of $0.75\%$ and $0.46\%$ over previous methods, respectively.

In summary, our contributions are threefold. First, we provide a PAC-Bayesian analysis of regularized fine-tuning. Our result implies empirical measures that explain the generalization performance of regularized fine-tuning. Second, we present a regularized self-labeling approach to enhance the generalization and robustness properties of fine-tuning. Third, we validate our approach on an extensive collection of classification tasks and pre-trained model architectures.

## 2 Related work

Fine-tuning is widely used in multi-task and transfer learning, meta-learning, and few-shot learning. Previous works (Li et al., 2018b,a) find that injecting $\ell_2$ regularization helps improve the performance of fine-tuning. Li et al. (2018b) propose a $\ell_2$ distance regularization method that penalizes the $\ell_2$ distance between the fine-tuned weights and the pre-trained weights. Li et al. (2018a) penalize the distance between the *feature maps* as opposed to the layer weights. Chen et al. (2019) design a regularization method that suppresses the spectral components (of the *feature maps*) with small

singular values to avoid negative transfer. Gouk et al. (2021) instead encode distance constraints in constrained minimization and use projected gradient descent to ensure the weights are close to the pre-trained model. Salman et al. (2020) show that fine-tuning from adversarially robust pre-trained models outperforms fine-tuning from (standard) pre-trained models.

The robustness of learning algorithms in the presence of label noise has been extensively studied in supervised learning (Natarajan et al., 2013). Three broad ideas for designing robust algorithms include defining novel losses, identifying noisy labels, and regularization methods. Zhang and Sabuncu (2018) design the robust Generalized Cross Entropy (GCE) loss which is a mixture of Cross Entropy (CE) and mean absolute error. The Symmetric Cross Entropy (SCE) (Wang et al., 2019) loss combines reverse cross-entropy with the CE loss. Ma et al. (2020) proposes APL, which normalizes the loss to be robust to noisy labels and combines active and passive loss functions. Thulasidasan et al. (2019) propose DAC, which identifies and suppresses the signals of noisy samples by abstention-based training. Liu et al. (2020) introduce ELR, an early learning regularization approach to mitigate label "memorization". Huang et al. (2020) propose a self-adaptive training method that corrects noisy labels and reweights training data to suppress erroneous signals. These works primarily concern the supervised learning setting. To the best of our knowledge, fine-tuning algorithms under label noise are relatively under-explored in transfer learning. Our approach draws inspiration from the semi-supervised learning literature, which has evaluated pseudo-labeling and self-training approaches given a limited amount of labeled data, and a large amount of unlabeled data (Zou et al., 2019; Xu et al., 2021; Tai et al., 2021). A very recent work considered a sharpness-aware approach and evaluated its performance for fine-tuning from noisy labels (Foret et al., 2021). It would be interesting to see if combining their approach with our ideas could lead to better results.

From a theoretical perspective, the seminal work of Ben-David et al. (2010) considers a setting where labeled data from many source tasks and a target task is available. They show that minimizing a weighted combination of the source and target empirical risks leads to the best result. In the supervised setting, Arora et al. (2019) provide data-dependent generalization bounds for neural networks. Nagarajan and Kolter (2018) and Wei and Ma (2019a,b) provide improved generalization bounds that depend only polynomially on the depth of the neural network characterized by a margin condition. Recent work has found that the PAC-Bayes theory provides generalization measures that correlate with empirical generalization performance better than other alternatives (Jiang et al., 2020). We defer a more extensive review of PAC-Bayesian generalization theory to Section A.

## 3 Preliminaries

**Problem setup.** We begin by formally introducing the setup. Suppose we would like to solve a target task. We have a training data set of size $n^{(t)}$. Let $(x_1^{(t)}, y_1^{(t)}), \ldots, (x_{n^{(t)}}^{(t)}, y_{n^{(t)}}^{(t)})$ be the feature vectors and the labels of the training data set. We assume that every $x_i^{(t)}$ lies in a $d$-dimensional space denoted by $\mathcal{X} \subset \mathbb{R}^d$. Following standard terminologies in statistical learning, we assume all data are drawn from some unknown distribution supported on $\mathcal{X} \times \mathcal{Y}$, where $\mathcal{Y} \subseteq \mathbb{R}$ is the label space. Denote the underlying data distribution as $\mathcal{P}^{(t)}$.

For our result in Section 4.1, we consider feedforward neural networks, and our results can also be extended to convolutional neural networks (e.g., using ideas from Long and Sedghi (2020)). Consider an $L$ layer neural network. For each layer $i$ from 1 to $L$, let $\psi_i$ be the activation function and let $W_i$ be the weight matrix at layer $i$. Given an input $z$ to the $i$-th layer, the output is then denoted as $\phi_i(z) = \psi_i(W_i z)$. Thus, the final output of the network is given by

$$f_W(x) = \phi_L \circ \phi_{L-1} \circ \cdots \phi_1(x), \text{ for any input } x \in \mathcal{X},$$

where we use $W = [W_1, \ldots, W_L]$ to include all the parameters of the network for ease of notation. The prediction error of $f_W$ is measured by a loss function $\ell(\cdot)$ that is both convex and 1-Lipschitz.

$$\mathcal{L}^{(t)}(f_W) = \mathop{\mathbb{E}}_{(x,y) \sim \mathcal{P}^{(t)}} \left[ \ell(f_W(x), y) \right]. \tag{1}$$

Suppose we have access to a pre-trained source model $\hat{W}^{(s)}$. Using $\hat{W}^{(s)}$ as an initialization, we then fine-tune the layers $\hat{W}_1^{(s)}, \ldots, \hat{W}_L^{(s)}$ to predict the target labels.

**Applications.** We consider seven image classification data sets described in Table 1. We use ResNets pre-trained on ImageNet as the initialization $\hat{W}^{(s)}$ (Russakovsky et al., 2015; He et al., 2016). We

Table 1: Basic statistics for seven image classification tasks.

| Datasets | Training | Validation | Test | Classes |
|---|---|---|---|---|
| Aircrafts (Maji et al., 2013) | 3334 | 3333 | 3333 | 100 |
| CUB-200-2011 (Wah et al., 2011) | 5395 | 599 | 5794 | 200 |
| Caltech-256 (Griffin et al., 2007) | 7680 | 5120 | 5120 | 256 |
| Stanford-Cars (Krause et al., 2013) | 7330 | 814 | 8441 | 196 |
| Stanford-Dogs (Khosla et al., 2011) | 10800 | 1200 | 8580 | 120 |
| Flowers (Nilsback and Zisserman, 2008) | 1020 | 1020 | 6149 | 102 |
| MIT-Indoor (Sharif Razavian et al., 2014) | 4824 | 536 | 1340 | 67 |

perform fine-tuning using the pre-trained network on the above data sets. See Section 5.1 for further description of the training procedure.

**Modeling label noise.** In many settings, fine-tuning is applied to a target task whose labels may contain noise; for example, if the target labels are created using weak supervision techniques (Ratner et al., 2019; Saab et al., 2021). To capture such settings, we denote a noisy data set as $(x_1^{(t)}, \tilde{y}_1^{(t)}), \ldots, (x_{n^{(t)}}^{(t)}, \tilde{y}_{n^{(t)}}^{(t)})$, where $\tilde{y}_i^{(t)}$ is a noisy version of $y_i^{(t)}$. We consider two types of label noise: *independent* noise and *correlated* noise. We say that the label noise is *independent* if it is independent of the input feature vector

$$\Pr(\tilde{y}_i^{(t)} = k | y_i^{(t)} = j, x_i^{(t)}) = \Pr(\tilde{y}_i^{(t)} = k | y_i^{(t)} = j) = \eta_{j,k}$$

for some fixed $\eta_{j,k}$ between 0 and 1. On the other hand, we say that the label noise is *correlated* if it depends on the input feature vector (i.e. the above equation does not hold).

## 4 Our proposed approaches

Given the problem setup described above, next, we study the generalization and robustness properties of regularized fine-tuning methods. First, we present a PAC-Bayes generalization bound for regularized fine-tuning. This result motivates us to evaluate two empirical measures: the fine-tuned distance in each layer and the perturbed loss of the fine-tuned model. Second, we consider fine-tuning from noisy labels. We show that layer-wise regularization prevents the model from memorizing the noisy labels. We then suggest injecting predictions of the model during training, similar to self-training and pseudo-labeling. Finally, we incorporate both components into our *regularized self-labeling* approach, blending the strengths of both to improve the generalization performance and the robustness of fine-tuning.

### 4.1 Fine-tuning and regularization

We consider the following regularized fine-tuning problem, which constrains the network from traveling too far from the pre-trained initialization $\hat{W}^{(s)}$ (Li et al., 2018a,b; Gouk et al., 2021).

$$\hat{W} \leftarrow \arg\min \hat{\mathcal{L}}^{(t)}(f_W) \tag{2}$$

$$\text{s.t. } \|W_i - \hat{W}_i^{(s)}\|_F \leq D_i, \ \forall i = 1, \ldots, L. \tag{3}$$

Above, $D_i$ is a hyperparameter that constrains how far the $i$-th layer $W_i$ can travel from the pre-trained initialization $\hat{W}_i^{(s)}$. Previous works have observed that stronger regularization reduces the generalization gap during fine-tuning (cf. Figure 2). Next, we analyze the generalization error of $\hat{W}$, that is, $\mathcal{L}^{(t)}(f_{\hat{W}}) - \hat{\mathcal{L}}^{(t)}(f_{\hat{W}})$, where $\mathcal{L}^{(t)}(f_{\hat{W}})$ is the test loss of $f_{\hat{W}}$ according to equation (1) and $\hat{\mathcal{L}}^{(t)}(f_{\hat{W}})$ is the empirical loss of $f_{\hat{W}}$ on the training data set.

**PAC-Bayesian analysis.** We begin by analyzing the generalization error of $f_{\hat{W}}$ using PAC-Bayesian

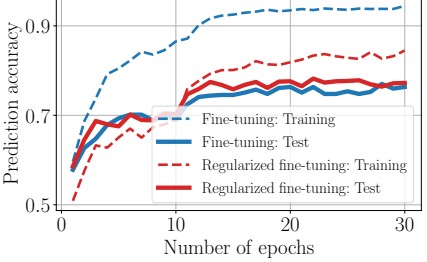

Figure 2: The training and test accuracy of fine-tuning using early stopping vs. optimizing (2) and (3) on the Indoor data set.

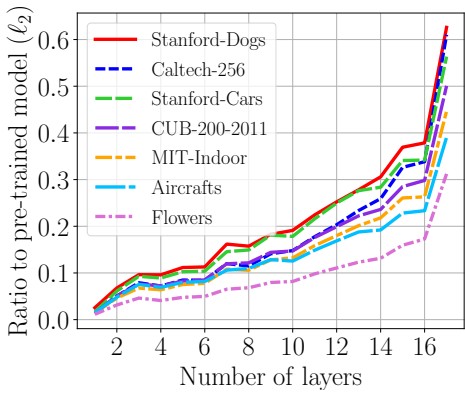

|  | CUB-200-2011 | | |
| $\sigma$ | Random | Pre-trained | Adversarial |
| --- | --- | --- | --- |
| $10^{-2}$ | $3.77\pm0.42$ | $\mathbf{1.45\pm0.13}$ | $1.76\pm0.09$ |
| $10^{-3}$ | $0.82\pm0.07$ | $0.62\pm0.03$ | $\mathbf{0.54\pm0.03}$ |
| $10^{-4}$ | $0.81\pm0.04$ | $0.61\pm0.03$ | $\mathbf{0.61\pm0.01}$ |
|  | Indoor | | |
| $\sigma$ | Random | Pre-trained | Adversarial |
| $10^{-2}$ | $2.51\pm0.34$ | $1.11\pm0.09$ | $\mathbf{0.97\pm0.07}$ |
| $10^{-3}$ | $0.49\pm0.09$ | $0.36\pm0.05$ | $\mathbf{0.32\pm0.04}$ |
| $10^{-4}$ | $0.44\pm0.03$ | $0.33\pm0.02$ | $\mathbf{0.30\pm0.04}$ |

(a) Fine-tuned distances.      (b) Perturbed loss.

Figure 3: Left: Ratio between the $\ell_2$-norm of the fine-tuned distances and the pre-trained network. Right: Perturbed loss of fine-tuned model from random initializations (Random), pre-trained model initializations (Pre-trained), adversarially trained model initializations (Adversarial). Results are shown for the CUB-200-2011 and the Indoor data sets using ResNet-18; Averaged over 10 runs.

tools (McAllester, 1999a,b). This departs from the previous work of Gouk et al. (2021), which bases on their analysis using Rademacher complexity. This change of perspective is inspired by several recent works that have found PAC-Bayesian bounds to better correlate with empirical performance than Rademacher complexity bounds (Jiang et al., 2020). We refer the interested reader to Section A for further references from this line of work. After presenting our result, we provide empirical measures of our result and discuss the practical implication.

**Theorem 4.1** (PAC-Bayes generalization bound for fine-tuning). *Suppose for every $i = 1, \ldots, L$, $\|\hat{W}_i^{(s)}\|_2 \leq B_i$, for a fixed $B_i > 1$. Suppose the feature vectors in the domain $\mathcal{X}$ are all bounded: $\|x\|_2 \leq C_1$ for every $x \in \mathcal{X}$, for some $C_1 \geq 1$. Finally, suppose the loss function $\ell(\cdot)$ is 1-Lipschitz and bounded from above by a fixed constant $C_2$. Under these conditions, let $f_{\hat{W}}$ be the minimizer of regularized fine-tuning, solved from problem (3). Let $\varepsilon > 0$ be an arbitrary small value. Let $H$ be the maximum over the width over all the $L$ layers and the input dimension $d$. Then, with probability at least $1 - 2\delta$ for some small $\delta > 0$, the expected loss $\mathcal{L}^{(t)}(f_{\hat{W}})$ is upper bounded by*

$$\mathcal{L}^{(t)}(f_{\hat{W}}) \leq \hat{\mathcal{L}}^{(t)}(f_{\hat{W}}) + \varepsilon + C_2 \sqrt{\frac{\frac{36}{\varepsilon^2}\cdot C_1^2 H \log(4LHC_2)\left(\sum_{i=1}^{L}\frac{\Pi_{j=1}^{L}(B_j+D_j)}{B_i+D_i}\right)^2\left(\sum_{i=1}^{L}D_i^2\right)+3\ln\frac{n^{(t)}}{\delta}+8}{n^{(t)}}}. \quad (4)$$

*Relation to prior works.* Compared to the result of Gouk et al. (2021), we instead proceed by factoring the generalization error into a noise error (i.e. the $\varepsilon$ term in equation (4)) and a KL-divergence between $\hat{W}^{(s)}$ and $\hat{W}$ (i.e. the final term in equation (4)). The proof of Theorem 4.1 is based on Neyshabur et al. (2018) and is presented in Appendix A. The difference between Theorem 4.1 and the result of Neyshabur et al. (2018) is that our result is stated for the (e.g. cross-entropy) loss function $\ell(\cdot)$ whereas Neyshabur et al. (2018) states the result using the soft margin loss.

Our result suggests that the *fine-tuned distances* $\{D_i\}_{i=1}^{L}$ and the *perturbed loss* (more precisely $\mathbb{E}_U\left[\ell(f_{\hat{W}+U}(x), y)\right]$ where every entry of $U$ is drawn from $\mathcal{N}(0, \sigma^2)$) are two important measures for fine-tuning. Next, we empirically evaluate these two measures on real-world data sets.

**Fine-tuned distances $\{D_i\}_{i=1}^{L}$.** First, we present our empirical finding for the fine-tuned distances in each layer. We fine-tune a ResNet-18 model (pre-trained on ImageNet) on seven data sets and calculate the Frobenius distance between $\hat{W}_i^{(s)}$ and $\hat{W}_i$ for every layer $i$.

Figure 3a shows the ratio of the fine-tuned distances to the pre-trained weights in each layer. First, we observe that the fine-tuned distances are relatively small compared to the pre-trained network. This means that fine-tuning stays within a small local region near the pre-trained model. Second, we find that $D_i$ varies across layers. $D_i$ is smaller at lower layers and is larger at higher layers. This observation aligns with the folklore intuition that different layers in Convolutional neural networks play a different role (Guo et al., 2019). Bottom layers extract higher-level representations of features

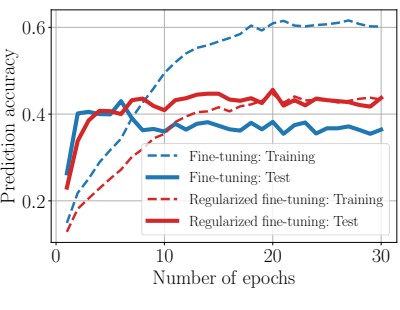 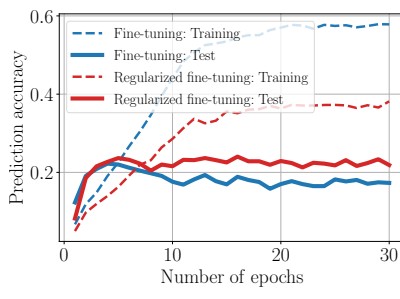

| (a) Noise rate $\eta = 60\%$. | (b) Noise rate $\eta = 80\%$. |

Figure 4: Training and test accuracy of fine-tuning using early stopping and optimizing equations (5) and (6) on the Indoor data set with different levels of noise rate. Stronger regularization reduces the generalization gap in both settings.

and stay close to the pre-trained model. Top layers extract class-specific features and vary among different tasks. Thus, top layers travel further away from the pre-trained model.

Based on the above empirical finding, we propose layer-wise constraints as regularization. In particular, the constraints for the bottom layers should be small, and the constraints for the top layers should be large. In practice, we find that an exponentially increasing scheme involving a base distance parameter $D$ and a scale factor $\gamma$ works well. This is summarized in the following regularized fine-tuning problem:

$$\hat{W} \leftarrow \arg\min \hat{\mathcal{L}}^{(t)}(f_W) \tag{5}$$

$$\text{s.t. } \|W_i - \hat{W}_i^{(s)}\|_F \leq D \cdot \gamma^{i-1}, \ \forall \, i = 1, \dots, L. \tag{6}$$

where the scale factor $\gamma > 1$ according to Figure 3a. The previous work of Gouk et al. (2021) uses the same distance parameter in equation (6), i.e., $D_i = D$, $\forall i = 1, \dots, L$. In our experiments, we have observed that such constant regularization constraints are only active at the top layers. With layer-wise regularization, we instead extend the constraints to the bottom layers as well. In Table 8 (cf. Section B.3), we compare the value of $\sum_{i=1}^{L} D_i^2$ between constant regularization and layer-wise regularization. We find that layer-wise regularization leads to a smaller value of $\sum_{i=1}^{L} D_i^2$, thus indicating a tighter generalization bound according to Theorem 4.1.

**Perturbed loss** $\mathbb{E}_U \left[ \ell(f_{\hat{W}+U}(x), y) \right]$. Second, we compare the perturbed loss of models fine-tuned from random, pre-trained, and adversarially robust pre-trained model initializations. The perturbed loss is measured by the average loss on the training set, after perturbing each entry of the layer weights by a Gaussian random variable $N(0, \sigma^2)$.

In Table 3b, we report the perturbed losses for both the CUB-200-2011 and the Indoor data set. The results show that models fine-tuned from pre-trained initializations are more stable than models fine-tuned from random initializations.

*Explaining why adversarial robustness helps fine-tuning.* Recent work (Salman et al., 2020) found that performing adversarial training in the pre-training phase leads to models that transfer better to downstream tasks. More precisely, the adversarial training objective is defined as

$$\tilde{\mathcal{L}}_{\text{adv}}^{(s)}(\theta) = \min_W \frac{1}{n^{(s)}} \sum_{i=1}^{n^{(s)}} \max_{\|\delta\|_2 \leq \varepsilon} \ell(f_W(x_i^{(s)} + \delta), y_i^{(s)}). \tag{7}$$

Let $\hat{W}_{\text{adv}}^{(s)}$ be a fine-tuned neural network using a pre-trained model from adversarial training. We measure the perturbed loss of $\hat{W}_{\text{adv}}^{(s)}$ using the pre-trained model provided by Salman et al. (2020) on the CUB-200-2011 and Indoor data set. Table 3b shows that $f_{\hat{W}_{\text{adv}}^{(s)}}$ often incurs lower perturbed losses than models fine-tuned from random and pre-trained initializations. In Section B.3, we additionally observe that the layers of $\hat{W}_{\text{adv}}^{(s)}$ generally have lower Frobenious norms compared to $\hat{W}^{(s)}$. Thus, the improved noise stability and the smaller layer-wise norms together imply a tighter generalization bound for adversarial training. We leave a thorough theoretical analysis of adversarial training in the context of fine-tuning to future works.

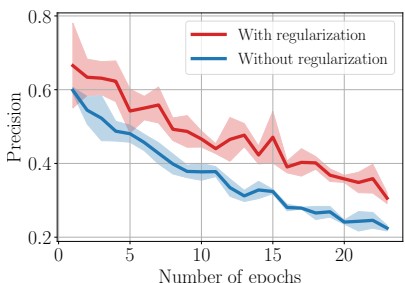

(a) Precision of self label-correction.

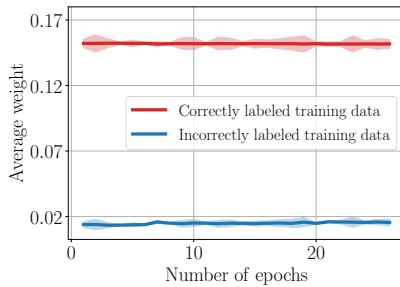

(b) Average weight injected by self label-reweighting.

Figure 5: Left: Comparing the precision of self-labeling with vs. without regularization. Precision is defined as the number of data points whose labels are corrected (cf. Line 7 in Alg. 1) divided by the number of data points whose labels are changed at every epoch. Right: Comparing the average normalized weight (cf. Line 10 in Alg. 1) of training data points with the correct label vs. data points with an incorrect label.

## 4.2  Fine-tuning and robustness

Next, we extend our approach to fine-tuning from noisy labels. We compare fine-tuning with and without regularization on a target task with independent label noise to motivate our approach. In Figure 4, we plot the training and test accuracy curves on the Indoor dataset with two noise rates. We observe that the test accuracy increases at first, meaning the network learns from the correct labels during the first few epochs. As the learning process progresses, the test accuracy of fine-tuning decreases, implying the network is "overfitting" to noisy labels. Furthermore, the gap between the training and test accuracy curves is more than 20%. Similar observations have been presented in prior works (Zhang et al., 2016; Li et al., 2018c; Liu et al., 2020), where over-parametrized neural networks can memorize the entire training set. One way to mitigate memorization is to explicitly reduce model capacity (Wu et al., 2020; Yang et al., 2021). Another way is via explicit regularization since the distance constraint significantly reduces the training-test accuracy gaps and improves test accuracy by a significant margin. Based on our empirical observation, the neural network has some discriminating power during the early fine-tuning phase. Thus, we can leverage the model predictions to relabel incorrect labels and reweight data points with incorrect labels. We describe the two components next.

**Self label-correction.** We propose a label correction step to augment the number of correct labels. We leverage the discriminating power of a network during the early phase and correct data points for which the model has high confidence. Concretely, let $p_i$ be the prediction of the $i$-th data point. Given a confidence threshold $p_t$ at epoch $t$, if $\max(p_i) \geq p_t$ and $\arg\max(p_i) \neq \tilde{y}_i^{(t)}$, we identify data point $i$ as noisy and relabel it to

$$\tilde{y}_i^{(t)} = \arg\max(p_i).$$

Figure 5a illustrates the precision of this step. We define precision as the number of correctly relabeled data points divided by the total number of data points whose labels are relabeled. We observe that precision is high at the beginning (around 60%) and gradually decreases in later epochs. Moreover, we find precision increases with regularization. This suggests an intricate interaction between regularization and self-labeling during fine-tuning. We elaborate on the role of regularization in label correction by looking at the number of relabeled data points in Section B.3.

**Self label-reweighting.** We incorporate a soft data removal step to prevent the model from overfitting to noisy labels. We identify data points as noisy if their loss values are large and thus down-weight them during training. The model will prioritize data points with correct labels in its gradient by assigning smaller weights to large-loss data points. More precisely, given a data point $(x_i^{(t)}, \tilde{y}_i^{(t)})$, we reweight it by $\omega_i = \exp(-\ell(f_W(x_i^{(t)}), \tilde{y}_i^{(t)})/\tau)$ where $\tau$ is a temperature parameter. In the implementation, we normalize the weights of every data point in every mini-batch $B$:

$$\hat{\mathcal{L}}_B(f_W) = \frac{1}{\sum_{i \in B} \omega_i} \sum_{i \in B} \omega_i \cdot \ell(f_W(x_i^{(t)}), \tilde{y}_i^{(t)}).$$

---

**Algorithm 1** Regularized self-labeling (REGSL)

---

**Input**: Input nosiy data $(x_1^{(t)}, \tilde{y}_1^{(t)}), \ldots, (x_{n^{(t)}}^{(t)}, \tilde{y}_{n^{(t)}}^{(t)})$, network model $f_W$, constraint distance $D$, distance scale factor $\gamma$, re-weight start step $E_r$, re-weight temperature $\tau$, label correction start step $E_c$, label correction threshold $p_t$, and fine-tuning steps $T$

1: Initialize model parameters with pre-trained weights $W = \hat{W}^s$ and the start step $t = 0$
2: **while** $t < T$ **do**
3:     Fetch a mini-batch of data $\{(x_i^{(t)}, \tilde{y}_i^{(t)})\}_{i=1}^m$
4:     **for** each data point $i$ in the batch **do**
5:         Calculate the predicted probability $\mathbf{p}_i = \text{softmax}(f_W(x_i^{(t)}))$
6:         **if** $t > E_c$ **and** $\max(\mathbf{p}_i) > p_t$ **and** $\arg\max(\mathbf{p}_i) \neq \tilde{y}_i^{(t)}$ **then**
7:           Correct the label $\tilde{y}_i^{(t)} = \arg\max(\mathbf{p}_i)$
8:         **end if**
9:         Calculate loss $\ell_i = \ell(f_W(x_i^{(t)}), \tilde{y}_i^{(t)})$
10:       **if** $t > E_r$ **then** Calculate the weight $\omega_i = \exp(-\ell_i/\tau)$ **else** Set the weight $\omega_i = 1$
11:     **end for**
12:     Update $f_W$ by SGD on $\mathcal{L} = -\frac{1}{\sum_i \omega_i} \sum_i^m \omega_i \ell_i$ and $t = t + 1$
13:     Project the weights $W_i$ of each layer inside the layer-wise constraint region as in Equation 6
14: **end while**

---

Figure 5b compares the average weight of data points with the correct label to data points with an incorrect label. We find that with our reweighting scheme, the average weight of incorrectly labeled data points is much lower than the average weight of correctly labeled data points. Thus, the reweighting scheme expands the gradient of correctly labeled data points, thus reducing the fraction of noisy data points in the training set.

The pseudo-code for our final approach, which combines both layer-wise regularization and self-labeling, is presented in Algorithm 1.

## 5 Experiments

We evaluate our proposed algorithm in a wide range of tasks and pre-trained networks. First, we show that our algorithm outperforms the baselines by $1.75\%$ on average over seven transfer learning tasks and can generalize to a more distant task of medical images. Second, our algorithm improves the robustness of fine-tuning in the presence of noisy labels compared to previous methods. Our algorithm improves over numerous baselines by $3.56\%$ on average under both independent and correlated label noise. Moreover, our approach can achieve a similar performance boost for fine-tuning vision transformer models on noisy labeled data. Finally, we show our algorithms improve over fine-tuning baselines by $0.75\%$ on average for the related task of few-shot classification. Our extensive evaluation confirms that our approach applies to a broad range of settings, validating our algorithmic insights. Due to space constraints, we defer the ablation studies to Section B.3. Our code is available at https://github.com/NEU-StatsML-Research/Regularized-Self-Labeling.

### 5.1 Experimental setup

**Data sets.** First, we evaluate fine-tuning on seven image classification data sets and one medical image data set. The statistics of the seven image data sets are described in Table 1. For the medical imaging task, we consider the ChestX-ray14 data set contains 112120 frontal-view chest X-ray images labeled with 14 different diseases (Wang et al., 2017; Rajpurkar et al., 2017). Second, we evaluate Algorithm 1 under two different kinds of label noise to test the robustness of fine-tuning. We selected the MIT-Indoor data set (Sharif Razavian et al., 2014) as the benchmark data set and randomly flipped the labels of the training samples. The results for the other data sets are similar, which can be found in Section B.2. We consider both independent random noise and correlated noise in our experiments. We generate the independent random noise by flipping the labels uniformly with a given noise rate. We simulate the correlated noise by using the predictions of an auxiliary network as noisy labels. Section B.1 describes the label flipping process. For few-shot image classification, we conduct experiments on the miniImageNet benchmark (Vinyals et al., 2016) following the setup of Tian et al. (2020).

Table 2: Top-1 test accuracy for fine-tuning ResNet-101 pre-trained on the ILSVRC-2012 subset of ImageNet. Results are averaged over 3 random seeds.

|  | Aircrafts | CUB-200-2011 | Caltech-256 | Stanford-Cars | Stanford-Dogs | Flowers | MIT-Indoor |
|---|---|---|---|---|---|---|---|
| Fine-tuning | 73.96±0.34 | 80.31±0.26 | 79.41±0.23 | 89.28±0.14 | 82.89±0.35 | 92.92±0.15 | 74.78±0.97 |
| $\ell^2$-Norm | 74.84±0.15 | 80.80±0.16 | 79.67±0.31 | 88.96±0.18 | 83.00±0.10 | 92.76±0.26 | 76.57±0.70 |
| LS | 74.19±0.41 | 82.22±0.17 | 81.10±0.13 | **89.66±0.10** | 84.14±0.21 | 93.72±0.04 | 76.84±0.31 |
| $\ell^2$-SP | 74.09±0.98 | 81.49±0.36 | 84.13±0.09 | 88.96±0.09 | 88.95±0.15 | 93.06±0.28 | 78.11±0.37 |
| $\ell^2$-PGM | 74.90±0.26 | 81.23±0.32 | 83.25±0.33 | 88.92±0.39 | 86.48±0.28 | 93.23±0.34 | 77.31±0.30 |
| REGSL (ours) | **75.32±0.23** | **82.24±0.21** | **84.90±0.16** | 89.14±0.22 | **89.58±0.13** | **93.82±0.35** | **79.30±0.31** |

**Architectures.** We use the ResNet-101 (He et al., 2016) network for transfer learning tasks and ResNet-18 (He et al., 2016) network for ChestX-ray data set. We use ResNet-18 network for the label-noise experiments and extend the our algorithm to the vision transformer (ViT) model (Dosovitskiy et al., 2020). The ResNet models are pre-trained on ImageNet (Russakovsky et al., 2015) and the ViT model is pre-trained on ImageNet-21k data set (Dosovitskiy et al., 2020). For the few-shot learning tasks, we use ResNet-12 pre-trained on the meta-training data set as in the previous work (Tian et al., 2020). We set four different values of $D_i$ for the four blocks of the ResNet models in our algorithm. We describe the fine-tuning procedure and the hyperparameters in Section B.1.

**Baselines.** For the transfer learning tasks, we use the Frobenius norm for distance regularization. We present ablation studies to compare the Frobenius norm and other norms in Section B.3. we compare our algorithm with fine-tuning with early stop (Fine-tuning), fine-tuning with weight decay ($\ell^2$-Norm), label smoothing (LS) method formulated in Müller et al. (2019), $\ell^2$-SP (Li et al., 2018b), and $\ell^2$-PGM (Gouk et al., 2021). For testing the robustness of our algorithm, in addition to the baselines described above, we adopt several baselines that have been proposed in the supervised learning setting, including GCE (Zhang and Sabuncu, 2018), SCE (Wang et al., 2019), DAC (Thulasidasan et al., 2019), APL (Ma et al., 2020), ELR (Liu et al., 2020) and self-adaptive training (SAT) (Huang et al., 2020). We compare with the same baselines to transfer learning in few-shot classification tasks. We describe the implementation and hyper-parameters of baselines in Section B.1.

## 5.2 Experimental results

**Improved regularization for transfer learning.** We report the test accuracy of fine-tuning ResNet-101 on seven data sets in Table 2. We observe that our method performs the best among six regularized fine-tuning methods on average. Our proposed layer-wise regularization method provides an improvement of $\mathbf{1.76}\%$ on average over distance-based regularization (Gouk et al., 2021). In particular, our approach outperforms $\ell^2$-PGM by $\mathbf{2 \sim 3}\%$ on both Stanford-Dogs and MIT-Indoor data sets. This suggests that adding appropriate distance constraints for all the layers is better than applying only one constraint to the top layers. In Table 10 (cf. Section B.3), we note that using the MARS norm of Gouk et al. (2021) yields comparable results to using the $\ell_2$ norm with our approach.

Next, we apply our approach to medical image classification, a more distant transfer task (relative to the source data set ImageNet). We report the mean AUROC (averaged over predicting all 14 labels) on the ChestX-ray14 data set in Table 6a (cf. Section B.2). With layer-wise regularization, we see an $\mathbf{0.39}\%$ improvement over the baseline methods. Finally, we extend our approach to text classification. The results are consistent with the above and can be found in Table 7 (cf. Section B.2).

**Improved robustness under label noise.** Next, we report the test accuracy of our approach on the Indoor data set with independent and correlated noise in Table 3. We find that our approach consistently outperforms baseline methods by $\mathbf{1 \sim 3}\%$ for various settings involving label noise. First, our method (REGSL) improves the performance by over $\mathbf{4}\%$ on average compared to distance-based regularization (Gouk et al., 2021). This result implies our method is more robust to label noise in the training labels. Second, our method outperforms previous supervised training methods by $\mathbf{3.56}\%$ on average. This result suggests that regularization is critical for fine-tuning. Taken together, our results suggest that regularization and self-labeling complement each other during fine-tuning. This is reinforced by our ablation study in Section B.3; We study the influence of each component of our algorithm and find that removing any component degrades performance.

We further apply our approach to fine-tuning pre-trained ViT (Dosovitskiy et al., 2020) from noisy labels, under the same setting as Table 3. Table 6b shows the result (cf. Section B.2). First, we find that

Table 3: Top-1 test accuracy of fine-tuning ResNet-18 on the Indoor data set with various settings of noisy labels in the training set. Results are averaged over 3 random seeds.

| Data sets | Methods | independent noise | | | | correlated noise |
|---|---|---|---|---|---|---|
| | | 20% | 40% | 60% | 80% | 25.18% |
| Indoor | Fine-tuning | 65.02±0.39 | 57.49±0.39 | 44.60±0.95 | 27.09±0.19 | 67.49±0.74 |
| | LS | 67.04±0.58 | 58.98±0.57 | 48.56±0.53 | 25.82±0.90 | 68.06±0.76 |
| | $\ell^2$-PGM | 69.45±0.19 | 62.74±0.60 | 51.24±0.15 | 30.15±0.94 | 68.86±0.27 |
| | GCE | 70.45±0.40 | 64.73±0.21 | 54.10±0.16 | 29.58±0.26 | 68.88±0.18 |
| | SCE | 69.43±0.20 | 64.68±0.45 | 55.07±0.52 | 29.85±0.44 | 69.00±1.22 |
| | DAC | 64.45±0.31 | 59.73±0.27 | 47.44±0.09 | 26.69±0.34 | 68.06±1.32 |
| | APL | 70.05±0.41 | 66.22±0.10 | 52.51±0.66 | 30.90±0.37 | 68.31±0.77 |
| | SAT | 68.98±0.63 | 63.43±0.72 | 52.84±0.38 | 29.60±0.53 | 67.06±0.55 |
| | ELR | 71.43±0.80 | 66.34±0.48 | 55.22±0.73 | 31.24±0.19 | 69.38±0.49 |
| | REGSL (ours) | **72.51±0.46** | **68.13±0.16** | **57.59±0.55** | **34.08±0.79** | **70.12±0.83** |

our approach improves upon the best regularization methods by **1.17**% averaged over two settings. Second, we find that our approach also improves upon self-labeling by **13.57**% averaged over two settings. These two results again highlight that both regularization and self-labeling contribute to the final result. While regularization prevents the model from overfitting to the random labels, self-labeling injects the belief of the fine-tuned model into the noisy data set.

**Extension to few-shot classification.** We extend our approach to a few-shot image classification task. We compare our approach to the baseline regularization methods. In Table 4, we report the average accuracy of 600 sampled tasks from the meta-test split of the miniImageNet benchmark (Vinyals et al., 2016). We find that our layer-wise regularization method achieves **0.75**% improvement on average compared to previous regularization methods. Additionally, regularization methods generally improve the performance of fine-tuning by over **1**%.

**Ablation studies.** We study the influence of removing each component from our algorithm. Results shown in Table 11 (cf. Section B.3) suggest that they all degrade performance. Thus, all three

Table 4: Test accuracy with 95% confidence interval for fine-tuning ResNet-12 over 600 meta-test splits of miniImageNet.

| Methods | miniImageNet | |
|---|---|---|
| | 5-way-1-shot | 5-way-5-shot |
| Fine-tuning | 60.56 ± 0.78 | 76.42 ± 0.36 |
| $\ell^2$-Norm | 60.93 ± 0.81 | 76.57 ± 0.55 |
| LS | 61.31 ± 0.77 | 76.73 ± 0.62 |
| $\ell^2$-SP | 61.48 ± 0.76 | 77.02 ± 0.62 |
| $\ell^2$-PGM | 61.35 ± 0.83 | 77.33 ± 0.59 |
| REGSL (ours) | **61.71 ± 0.77** | **78.03 ± 0.54** |

components in Algorithm 1 contribute to the final performance. The importance of each component depends on the noise rate. In particular, if the noise rate is higher than 40%, label-correction is the most critical component. Other ablation studies we present include: comparing the $\ell_2$ norm and the MARS norm in our algorithm and comparing the distance parameters between layer-wise and constant regularization. We leave the details to Section B.3.

## 6 Conclusion

This paper studied regularization methods for fine-tuning as well as their robustness properties. We investigated the generalization error of fine-tuning using PAC-Bayesian techniques. This leads to two empirical measures, which we empirically computed. The analysis inspired us to consider layer-wise regularization for fine-tuning. This approach performs well on both pre-trained ResNets and ViTs; we have found the fine-tuned distance using vanilla fine-tuning and encoded the distance patterns in layer-wise regularization. We then evaluated the performance of regularized fine-tuning from noisy labels. We proposed a self label-correction and label-reweighting approach in the noisy setting. We found that regularization and self-labeling complement each other in our experiments, leading to significant improvement over previous methods, which either use regularization methods or use self-labeling methods but not both. We discuss the limitation of our work in Section C.

**Negative societal impacts.** Our work concerns the theory and the empirical performance of fine-tuning for transfer learning. Due to its technical nature, there is very little negative societal impact. Since transfer learning is commonly used in practice, one avenue that deserves attention for these applications is to carefully evaluate bias or fairness metrics before applying our proposed approaches.

**Acknowledgment.** Thanks to the anonymous reviewers and the area chairs for carefully reading through our work and providing constructive feedback. Thanks to the program committee chairs for communicating with us regarding an omitted factor in equation (14), which we have added in the current version. HZ is grateful to Sen Wu and Chris Ré for stimulating discussions. HZ is supported by Northeastern University's Khoury seed/proof-of-concept grant.

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
