# A Proof of Theorem 4.1

**Background.** PAC-Bayesian generalization theory offers a nice way to blend data-dependent properties such as noise stability and sharpness into generalization bounds. Recent works (e.g., Bartlett et al. (2017); Neyshabur et al. (2018)) introduced generalization bounds for multi-layer neural networks in an attempt to explain why neural network models generalize well despite having more trainable parameters than the number of training examples. On the one hand, the VC dimension of a neural network is known to be roughly equal to its number of parameters (Bartlett et al., 2019). On the other hand, the number of parameters is not a good capacity measure for neural nets, as evidenced by the popular work of Zhang et al. (2016). The bounds of Bartlett et al. (2017) (and follow-up works) provide a more meaningful notion of "capacity" compared to VC-dimension.

While these bounds constitute an improvement compared to classical learning theory, it is unclear if these bounds are tight or non-vacuous. One response to this criticism is a computational framework from Dziugaite and Roy (2017). That work shows that directly optimizing the PAC-Bayes bound leads to a much smaller bound and low test error simultaneously (see also Zhou et al. (2018) for a large-scale study). The recent work of Jiang et al. (2020) further compared different complexity notions and noted that the ones given by PAC-Bayes tools correlate better with empirical performance. In particular, we will use the following classical result (McAllester, 1999a,b).

**Theorem A.1** (Theorem 1 in McAllester (1999a)). *Let $\mathcal{H}$ be some hypothesis class. Let $P$ be a prior distribution on $\mathcal{H}$ that is independent of the training set. Let $Q_S$ be a posterior distribution on $\mathcal{H}$ that may depend on the training set $S$. Suppose the loss function is bounded from above by $C$. With probability $1 - \delta$ over the randomness of the training set, the following holds*

$$\mathbb{E}_{h \sim Q_S} [\mathcal{L}(h)] \leq \mathbb{E}_{h \sim Q_S} \left[ \hat{\mathcal{L}}(h) \right] + C \sqrt{\frac{\mathrm{KL}(Q_S \| P) + 3 \ln \frac{n}{\delta} + 8}{n}}. \tag{8}$$

We remark that the original statement in McAllester (1999a) requires the loss function is bounded between $0$ and $1$. The above statement modifies the original statement and instead applies to a loss function bounded between $0$ and $C$, for some fixed constant $C > 0$. This is achieved by rescaling the loss by $1/C$, leading the $C$ factor in the right hand side of equation (8). To invoke the above result in our setting, we set the prior distribution $P = \mathcal{N}(\hat{W}^{(s)}, \sigma^2 \,\mathrm{Id})$, where $\hat{W}^{(s)}$ are the weights of the pre-trained nework. The posterior distribution $Q_S$ is centered at the fine-tuned model as $\mathcal{N}(\hat{W}, \sigma^2 \,\mathrm{Id})$. Based on the above result, we present the proof of Theorem 4.1.

*Proof of Theorem 4.1.* First, we show that the KL divergence between $P$ and $Q_S$ is equal to $\frac{1}{2\sigma^2} \| \hat{W} - \hat{W}^{(s)} \|^2$. We expand the definition using the density of multivariate normal distributions.

$$
\begin{aligned}
\mathrm{KL}(P \| Q_S) &= \mathbb{E}_{W \sim \mathcal{P}} \left[ \log \left( \frac{\mathrm{Pr}(W \sim P)}{\mathrm{Pr}(W \sim Q_S)} \right) \right] \\
&= \mathbb{E}_{W \sim \mathcal{P}} \left[ \log \frac{\exp(-\frac{1}{2\sigma^2} \| W - \hat{W}^{(s)} \|^2)}{\exp(-\frac{1}{2\sigma^2} \| W - \hat{W} \|^2)} \right] \\
&= -\frac{1}{2\sigma^2} \mathbb{E}_{W \sim \mathcal{P}} \left[ \| W - \hat{W}^{(s)} \|^2 - \| W - \hat{W} \|^2 \right] \\
&= \frac{1}{2\sigma^2} \mathbb{E}_{W \sim \mathcal{P}} \left[ \langle \hat{W}^{(s)} - \hat{W}, 2W - \hat{W}^{(s)} - \hat{W} \rangle \right] \\
&= \frac{1}{2\sigma^2} \| \hat{W} - \hat{W}^{(s)} \|_F^2 \leq \frac{\sum_{i=1}^{L} D_i^2}{2\sigma^2},
\end{aligned}
$$

where the last line uses the fact that $\| \hat{W}_i - \hat{W}_i^{(s)} \|_F \leq D_i$, for all $1 \leq i \leq L$.

Next, let the dimension of $W_i$ be $d_{i-1}$ times $d_i$, for every $i = 1, 2, \ldots, L$. In particular, $d_0$ is equal to the dimension of the input data points. Let $H = \max_{0 \leq i \leq L} d_i$ be the maximum matrix dimension size across all the $L$ layers. Let $e = \sigma \sqrt{2H \log(2L \cdot H/\delta)}$. We show that for any $\delta > 0$ and any $L$-layer feedforward neural network parameterized by $W = [W_1, W_2, \ldots, W_L]$, with probability at least $1 - \delta$, the perturbation $Q_S$ increases the loss function $\hat{\mathcal{L}}(h)$ by at most

$$C_1 \cdot e \cdot \left( \sum_{i=1}^{L} \frac{\prod_{j=1}^{L} (\|W_j\|_2 + e)}{\|W_i\|_2 + e} \right). \tag{9}$$

Consider some data point $(x, y)$ evaluated at a neural network function $f_W$. Let $U_i$ be a Gaussian perturbation matrix on $W_i$ with mean zero and entrywise variance $\sigma^2$, for $i = 1, \ldots, L$. Denote by $U = [U_1, U_2, \ldots, U_L]$. Since $\ell(x, y)$ is 1-Lipschitz over $x$, we get

$$|\ell(f_{W+U}(x), y) - \ell(f_W(x), y)| \leq \|f_{W+U}(x) - f_W(x)\|_2. \tag{10}$$

Since $U_i \in \mathbb{R}^{d_{i-1} \times d_i}$ is a random matrix with i.i.d. entries sampled from the standard Gaussian distribution, we can upper bound the operator norm of $U_i$ by applying well-known concentration results. In particular, by equation (4.1.8) and section (4.2.2) of Tropp (2015), we get

$$\Pr\left[\frac{1}{\sigma}\|U_i\|_2 \geq t\right] \leq (d_{i-1} + d_i) \cdot \exp\left(-\frac{-t^2}{2\max(d_{i-1}, d_i)}\right). \tag{11}$$

Thus, for all $i = 1, 2, \ldots, L$, with probability at most $\delta/L$, we have that

$$\frac{1}{\sigma}\|U_i\|_2 \leq \sqrt{2H \log(2L \cdot H/\delta)}. \tag{12}$$

In particular, the right hand side above is obtained by setting $2H \exp\left(-\frac{t^2}{2H}\right) = \delta/L$ and solve for $t = \sqrt{2H \log(2L \cdot H/\delta)}$. Since the right hand side of equation (11) is at most $2H \exp\left(-\frac{t^2}{2H}\right)$, by union bound, we conclude that with probability at most $1 - \delta$, for all $i = 1, 2, \ldots, L$, the operator norm of $U_i$ is at most $\sigma \cdot \sqrt{2H \log(2L \cdot H/\delta)}$.

Let $z_i$ be the input to the $i$-th layer of $f_W$. Let $\tilde{z}_i$ be the input to the $i$-th layer of $f_{W+U}$. We can expand the right hand side of equation (10) as

$$\begin{aligned}
&\|\psi_L((W_L + U_L)\tilde{z}_L) - \psi_L(W_L z_L)\|_2 &&(13)\\
&\leq \|(W_L + U_L)\tilde{z}_L - W_L z_L\|_2 &&(\text{since } \psi_L(\cdot) \text{ is 1-Lipschitz})\\
&\leq \|W_L(\tilde{z}_L - z_L)\|_2 + \|U_L \tilde{z}_L\|_2 &&(\text{by triangle inequality})\\
&\leq \|W_L\|_2 \cdot \|\tilde{z}_L - z_L\|_2 + \|U_L \tilde{z}_L\|_2 &&(\|Wx\|_2 \leq \|W\|_2 \cdot \|x\|_2 \text{ for any vector } x)\\
&\leq \|W_L\|_2 \cdot \|\tilde{z}_L - z_L\|_2 + \sigma \cdot \sqrt{2H \log(2L \cdot H/\delta)} \cdot \|\tilde{z}_L\|_2, &&(14)
\end{aligned}$$

where the last step is by equation (12). Recall from Section 3 that the $i$-th layer takes an input $z$ and outputs $\psi_i(W_i z)$, for every $i = 1, \ldots, L$. Because we have assumed that $\psi_i(\cdot)$ is 1-Lipschitz, for all $i = 1, \ldots, L-1$, we can bound

$$\begin{aligned}
\|\tilde{z}_L\|_2 = \phi_{L-1} \circ \phi_{L-2} \cdots \circ \phi_1(x) &\leq \left(\prod_{i=1}^{L-1} \|W_i + U_i\|_2\right) \cdot \|x\|_2\\
&\leq \left(\prod_{i=1}^{L-1} \|W_i + U_i\|_2\right) \cdot C_1 \quad (\text{since } \|x\|_2 \leq C_1 \text{ for } x \in \mathcal{X})\\
&\leq \left(\prod_{i=1}^{L-1} (\|W_i\|_2 + e)\right) \cdot C_1. \quad (\text{by equation (12)})
\end{aligned}$$

Applying the above to equation (14), we have shown

$$\|f_{W+U}(x) - f_W(x)\|_2 \leq \|W_L\|_2 \cdot \|\tilde{z}_L - z_L\|_2 + \sigma \cdot \sqrt{2H \log(2L \cdot H/\delta)} \cdot \left(\prod_{i=1}^{L-1}(\|W_i\|_2 + e)\right) \cdot C_1. \tag{15}$$

Next, we can expand the difference between $\tilde{z}_L = \psi_{L-1}((W_{L-1} + U_{L-1})\tilde{z}_{L-1})$ and $z_L = \psi_{L-1}(W_{L-1} z_{L-1})$ similar to equation (13). In general, for any $i = 1, 2, \ldots, L$, we get

$$\|\tilde{z}_i - z_i\|_2 \leq \|W_{i-1}\|_2 \cdot \|\tilde{z}_{i-1} - z_{i-1}\|_2 + \sigma \cdot \sqrt{2H \log(2L \cdot H/\delta)} \cdot \left(\prod_{i=1}^{i-2}(\|W_i\|_2 + e)\right) \cdot C_1. \tag{16}$$

By repeatedly applying equation (16) together with equation (15) (relaxing $\|W_i\|_2$ to $\|W_i\|_2 + e$), we can show that with probability at least $1 - \delta$,

$$\|f_{W+U}(x) - f_W(x)\|_2 \leq \sigma \cdot C_1 \sqrt{2H \log(2L \cdot H/\delta)} \cdot \left(\sum_{i=1}^{L} \frac{\prod_{j=1}^{L}(\|W_j\|_2 + e)}{\|W_i\|_2 + e}\right).$$

Combined with equation (10), we have shown that equation (9) holds. Finally, with equation (9) and the fact that that the loss function $\ell(\cdot)$ is bounded from above by some fixed value $C_2$, we can bound the expectation of $\hat{\mathcal{L}}(h)$ for some $h \sim Q_S$ as

$$\mathbb{E}_{h \sim Q_S}\left[\hat{\mathcal{L}}(h)\right] \leq \hat{\mathcal{L}}(f_{\hat{W}}) + (1-\delta) \cdot \sigma \cdot C_1 \sqrt{2H \log\left(\frac{2L \cdot H}{\delta}\right)} \cdot \left(\sum_{i=1}^{L} \frac{\prod_{j=1}^{L}(\|\hat{W}_j\|_2 + e)}{\|\hat{W}_i\|_2 + e}\right) + \delta \cdot C_2, \quad (17)$$

Note that the above inequality applies for any (small) value of $\delta$. Thus, we can minimize the right hand side by appropriately setting $\delta$. For our purpose, we will show that there exists some $\delta$ such that the right hand side of equation (17) (leaving out $\hat{\mathcal{L}}(f_{\hat{W}})$) is at most

$$2\sigma C_1 \left(\sum_{i=1}^{L} \frac{\prod_{j=1}^{L}(\|\hat{W}_j\|_2 + e)}{\|\hat{W}_i\|_2 + e}\right) \cdot \sqrt{2H \log\left(4L \cdot H \cdot C_2\right)}. \quad (18)$$

To see that the above is true, we will abstract away the precise scalars and instead write the right hand side of equation (17) as $g(\delta) = (1-\delta)A_1\sqrt{\log(A_2/\delta)} + C_2\delta$, for some fixed $A_1$ and $A_2$ that does not depend on $\delta$. We note that subject to

$$C_2\delta \leq (1-\delta)A_1\sqrt{\log(A_2/\delta)}, \quad (19)$$

$g(\delta)$ is at most

$$g(\delta) \leq 2(1-\delta)A_1\sqrt{\log(A_2/\delta)} \leq 2A_1\sqrt{\log\left(\frac{A_2}{\delta}\right)}. \quad (20)$$

Therefore, we only need to lower bound $\delta$ based on the constraint (19). In particular, the largest possible $\delta^\star$ under constraint (19) is achieved when both sides equal:

$$C_2\delta^\star = (1-\delta^\star)A_1\sqrt{\log(A_2/\delta^\star)} \geq \frac{A_1}{2}\sqrt{\log(A_2)},$$

which implies that $\delta^\star \geq \frac{A_1}{2C_2}\sqrt{\log(A_2)}$. Thus, we have shown that $g(\delta^\star)$ is less than equation (18), using equation (20) and the lower bound on $\delta^\star$. Additionally, $A_1$ is on the order of a fixed constant based on $\sigma$ defined below.

Using a similar argument since our bound on the perturbed loss holds point-wise for every $x \in \mathcal{X}$, we can likewise prove that with probability $1-\delta$,

$$\mathbb{E}_{h \sim Q_S}[\mathcal{L}(h)] \geq \mathcal{L}(f_{\hat{W}}) - 2\sigma C_1 \left(\sum_{i=1}^{L} \frac{\prod_{j=1}^{L}(\|\hat{W}_j\|_2 + e)}{\|\hat{W}_i\|_2 + e}\right) \cdot \sqrt{2H \log(4L \cdot H \cdot C_2)}.$$

By applying equation (18) and the above to equation (8) in Theorem A.1, we thus conclude that for any $\sigma$, with probability at least $1 - 2\delta$ over the training data set, the following holds

$$\mathcal{L}(f_{\hat{W}}) \leq \hat{\mathcal{L}}(f_{\hat{W}}) + 4\sigma \cdot C_1 \cdot \sqrt{2H \log(4L \cdot H \cdot C_2)} \left(\sum_{i=1}^{L} \frac{\prod_{j=1}^{L}(\|\hat{W}_j\|_2 + e)}{\|\hat{W}_i\|_2 + e}\right) + C_2 \sqrt{\frac{\sum_{i=1}^{L} \frac{D_i^2}{2\sigma^2} + 3\ln\frac{n^{(t)}}{\delta} + 8}{n^{(t)}}}. \quad (21)$$

Since $\|\hat{W}_i^{(s)}\|_2 \leq B_i$ and $\|\hat{W}_i - \hat{W}_i^{(s)}\|_F \leq D_i$, for any $i = 1, \ldots, L$, we have that $\|\hat{W}_i\|_2 \leq B_i + D_i$. Thus, we can upper bound the above by replacing every $\|\hat{W}_i\|_2$ with $B_i + D_i$.

By setting $\sigma = \frac{\varepsilon}{6\sigma C_1 \cdot \alpha \sqrt{2H \log(4LHC_2)}}$ where $\alpha = \left(\sum_{i=1}^{L} \frac{\prod_{j=1}^{L}(B_j + D_j)}{B_i + D_i}\right)$, we get $e \leq \frac{1}{6\alpha}$ since $C_1 \geq 1$. To finish the proof, we will show

$$\sum_{i=1}^{L} \frac{\prod_{j=1}^{L}(B_j + D_j + e)}{B_i + D_i + e} \leq 3\alpha/2.$$

In particular, we will show that for every $k = 1, 2, \ldots, L$, $e \leq \frac{B_k + D_k}{6L}$. To see this, recall that $B_k \geq 1$ and $D_k \geq 0$. Therefore, $\alpha \geq L$ for every $k$ and $e \leq \frac{1}{6\alpha} \leq \frac{B_k + D_k}{6L}$. This implies

$$\sum_{i=1}^{L} \frac{\prod_{j=1}^{L}(B_j + D_j + e)}{(B_i + D_i + e)} \leq \left(1 + \frac{1}{6L}\right)^{L-1} \sum_{i=1}^{L} \frac{\prod_{j=1}^{L}(B_j + D_j)}{B_i + D_i} \leq \frac{3}{2}\alpha.$$

Table 5: Top-1 test accuracy on CUB-200-2011 and Flowers dataset with independent label noise injected in the training set. Results are averaged over 3 random seeds.

| Datasets | Methods | independent noise | | | |
|---|---|---|---|---|---|
| | | 20% | 40% | 60% | 80% |
| CUB-200-2011 | Fine-tuning | $68.28 \pm 0.34$ | $56.88 \pm 0.38$ | $39.54 \pm 0.42$ | $16.21 \pm 0.38$ |
| | LS | $69.89 \pm 0.85$ | $59.18 \pm 0.43$ | $41.58 \pm 0.45$ | $16.96 \pm 0.45$ |
| | $\ell^2$-PGM | $69.71 \pm 0.31$ | $58.91 \pm 0.42$ | $41.52 \pm 0.74$ | $16.76 \pm 0.41$ |
| | GCE | $69.54 \pm 0.25$ | $60.15 \pm 0.93$ | $41.84 \pm 0.47$ | $17.77 \pm 0.32$ |
| | SCE | $70.68 \pm 0.67$ | $61.33 \pm 0.57$ | $43.67 \pm 0.37$ | $17.62 \pm 0.19$ |
| | DAC | $68.58 \pm 0.25$ | $57.37 \pm 0.29$ | $40.00 \pm 0.37$ | $15.92 \pm 0.83$ |
| | APL | $70.59 \pm 0.18$ | $59.68 \pm 0.69$ | $40.46 \pm 0.24$ | $14.69 \pm 0.36$ |
| | SAT | $68.69 \pm 0.38$ | $57.34 \pm 0.18$ | $38.75 \pm 0.40$ | $15.14 \pm 0.20$ |
| | ELR | $69.92 \pm 0.14$ | $58.69 \pm 0.02$ | $40.76 \pm 0.68$ | $16.68 \pm 0.79$ |
| | REGSL (ours) | $\mathbf{71.76 \pm 0.49}$ | $\mathbf{62.79 \pm 0.23}$ | $\mathbf{45.85 \pm 0.66}$ | $\mathbf{17.88 \pm 0.25}$ |
| Flowers | Fine-tuning | $83.13 \pm 0.15$ | $72.23 \pm 0.40$ | $55.27 \pm 0.32$ | $29.35 \pm 0.74$ |
| | LS | $83.62 \pm 0.30$ | $72.35 \pm 0.47$ | $54.23 \pm 0.24$ | $28.60 \pm 0.45$ |
| | $\ell^2$-PGM | $83.45 \pm 0.70$ | $73.24 \pm 0.20$ | $56.51 \pm 0.40$ | $31.04 \pm 0.88$ |
| | GCE | $83.09 \pm 0.22$ | $71.74 \pm 0.65$ | $54.73 \pm 0.90$ | $28.38 \pm 0.95$ |
| | SCE | $83.45 \pm 0.14$ | $73.11 \pm 0.05$ | $55.96 \pm 0.97$ | $29.22 \pm 0.07$ |
| | DAC | $83.40 \pm 0.27$ | $72.73 \pm 0.34$ | $55.27 \pm 0.40$ | $28.95 \pm 0.89$ |
| | APL | $83.42 \pm 0.11$ | $72.06 \pm 0.23$ | $54.96 \pm 0.36$ | $28.86 \pm 0.64$ |
| | SAT | $82.49 \pm 0.25$ | $72.52 \pm 0.28$ | $55.10 \pm 0.50$ | $27.82 \pm 0.38$ |
| | ELR | $83.38 \pm 0.38$ | $72.53 \pm 0.48$ | $55.74 \pm 0.42$ | $30.17 \pm 0.26$ |
| | REGSL (ours) | $\mathbf{83.79 \pm 0.39}$ | $\mathbf{73.32 \pm 0.66}$ | $\mathbf{57.52 \pm 0.15}$ | $\mathbf{31.09 \pm 0.25}$ |

Thus, plugging in the value of $\sigma$ above, we conclude that the second term in equation (21) is at most $\varepsilon$. By applying the value of $\sigma$ to the third term in equation (21), we conclude

$$\mathcal{L}(f_{\hat{W}}) \leq \hat{\mathcal{L}}(f_{\hat{W}}) + \varepsilon + C_2 \sqrt{\frac{\frac{36}{\varepsilon^2} \cdot C_1^2 H \log(4LHC_2) \left( \sum_{i=1}^{L} \frac{\prod_{j=1}^{L}(B_j + D_j)}{B_i + D_i} \right)^2 \left( \sum_{i=1}^{L} D_i^2 \right) + 3 \ln \frac{n^{(t)}}{\delta} + 8}{n^{(t)}}}.$$

Thus, we have proved equation (4) is true. The proof is complete. □

*Implication.* Our result implies a tradeoff in setting the distance parameter $D_i$. While a larger $D_i$ increases the network's "capacity", the generalization bound gets worse as a result.

## B  Extended experimental setup and results

### B.1  Additional details of the experimental setup

We extend the details of our experiment setup. First, we introduce the data sets in our experiments and describe the pre-processing steps. Second, we present the model architectures and the fine-tuning procedures. Third, we describe the baselines in detail. Finally, we describe the implementation and the hyperparameters for our algorithm and the baselines.

**Data sets.** We evaluate fine-tuning on seven image classification data sets covering multiple applications, including fine-grained object recognition, scene recognition, and general object recognition. We divide each dataset into the training set, the validation set, and the test set for data split. We adopt the standard splitting of data given in the Aircrafts dataset (Maji et al., 2013). For Caltech-256 dataset (Griffin et al., 2007), we use the setting in Li et al. (2018b) that randomly samples 30, 20, 20 images of each class for training, validation, and test set, respectively. For other data sets, we split 10% of the training set as the validation set and use the standard test set. We describe the statistics of the seven data sets in Table 1. When fine-tuning on the seven data sets, we preserve the original pixel and resize the shorter side to 256 pixels. The image samples are normalized with the mean and std values over ImageNet data (Russakovsky et al., 2015). Moreover, we apply commonly used

Table 6: Left: Mean AUROC of fine-tuning ResNet-18 on the ChestX-ray14 data set. Right: Top-1 accuracy of fine-tuning ViT on the indoor data set. Results are averaged over 3 random seeds.

<table>
<tr><td colspan="2" align="center">(a) The ChestX-ray14 data set.</td></tr>
<tr><td>Methods</td><td>Mean AUROC</td></tr>
<tr><td>Fine-tuning</td><td>$0.8159 \pm 0.0667$</td></tr>
<tr><td>$\ell^2$-Norm</td><td>$0.8198 \pm 0.0644$</td></tr>
<tr><td>LS</td><td>$0.7885 \pm 0.0578$</td></tr>
<tr><td>$\ell^2$-SP</td><td>$0.8231 \pm 0.0658$</td></tr>
<tr><td>$\ell^2$-PGM</td><td>$0.8235 \pm 0.0636$</td></tr>
<tr><td>REGSL (ours)</td><td>$\mathbf{0.8274 \pm 0.0654}$</td></tr>
</table>

(b) The indoor data set.

| Methods | Independent Noise | |
| --- | --- | --- |
| | 40% | 80% |
| Fine-tuning | $75.87 \pm 1.15$ | $32.06 \pm 3.17$ |
| Regularization | $82.66 \pm 0.44$ | $63.01 \pm 0.83$ |
| Self-labeling | $79.13 \pm 0.37$ | $41.69 \pm 0.16$ |
| REGSL (ours) | $\mathbf{83.48 \pm 0.29}$ | $\mathbf{64.50 \pm 0.53}$ |

data augmentation methods on the training samples, including random scale cropping and random flipping. The images are resized to $224 \times 224$ as the input for the model. The training and batch size are both 16.

To test the robustness of fine-tuning, we evaluate Algorithm 1 under two different settings with label noise. We select MIT-Indoor dataset (Sharif Razavian et al., 2014) as the benchmark dataset and randomly flipped the labels of the training samples. The results for the other datasets are similar, which can be found in Appendix B.2. We consider two scenarios of label noise in our experiments, independent random noise and correlated noise. Random label noise is generated by flipping the labels of a given proportion of training samples to other classes uniformly. For the correlated noise setting, the label noise is dependent on the sample. We simulate the correlated noisy label by training an auxiliary network on a held-out dataset to a certain accuracy. We then use the prediction of the auxiliary network as noisy labels. For few-shot learning, we conduct experiments on a few-shot image recognition benchmark, miniImageNet (Vinyals et al., 2016) and follow the meta training and meta test setup in Tian et al. (2020).

**Architectures.** For image data sets, we use ResNet-101 (He et al., 2016) network which is pre-trained on ImageNet dataset (Russakovsky et al., 2015). For the label-noise experiments, we mainly use ResNet-18 (He et al., 2016) network. In the transfer learning and label-noise experiments, we fine-tune the model with Adam optimizer (Kingma and Ba, 2014) with an initial learning rate 0.0001 for 30 epochs and decay the learning rate by 0.1 every 10 epochs. We report the average Top-1 accuracy on the test set of 3 random seeds. For the few-shot learning tasks, we use ResNet-12 pre-trained on the meta-training dataset as in the previous work (Tian et al., 2020). To fine-tune on the meta-test sets, we use Adam optimizer (Kingma and Ba, 2014) with an initial learning rate $5e^{-5}$ and fine-tune the model on the training set for 25 epochs. We report the average classification accuracies on 600 sampled tasks from the meta-test split.

**Baselines.** For the transfer learning tasks, we focus on using the Frobenius norm for distance regularization. We present ablation studies to compare the Frobenius norm and other norms in Appendix B.2. we compare our algorithm with fine-tuning with early stop (Fine-tuning), fine-tuning with weight decay ($\ell^2$-Norm), label smoothing (LS) method formulated in (Müller et al., 2019), $\ell^2$-SP (Li et al., 2018b), and $\ell^2$-PGM (Gouk et al., 2021). For testing the robustness of our algorithm, in addition to the baselines described above, we adopt several baselines that have been proposed in the supervised learning setting, including GCE (Zhang and Sabuncu, 2018), SCE (Wang et al., 2019), DAC (Thulasidasan et al., 2019), APL (Ma et al., 2020), ELR (Liu et al., 2020) and self-adaptive training (SAT) (Huang et al., 2020). We compare with the same fine-tuning baselines in transfer learning for the few-shot classification tasks.

**Implementation and hyperparameters.** For all baselines and data sets, all regularization hyperparameters are searched on the validation dataset by the Optuna optimization framework (Akiba et al., 2019). In our proposed algorithm, we search the constraint distance $D$ in $[0.05, 10]$ and the distance scale factor $\gamma$ in $[1, 5]$ by sampling. For setting layer-wise constraints, we set four different constraints for the four blocks of the ResNet. Concretely, ResNet-101 has four blocks, with 10, 13, 67, and 10 convolutional layers for each block, respectively. For every block, we set the same constraint value for all the layers within the block. Therefore, there are four distance values $(D, D \cdot \gamma, D \cdot \gamma^2, D \cdot \gamma^3)$ each block. Similar constraints are set for ResNet-18 and ResNet-12 models in the experiments. The search space for other hyperparameters is shown as follows. Re-weight start step $E_r$ is searched in $\{3, 5, 8, 10, 13\}$. Re-weight temperature factor $\gamma$ is searched in $\{3.0, 2.0, 1.5, 1.0\}$. Label correction

Table 7: Top-1 accuracy of fine-tuning a three-layer feed forward network pre-trained on SST with no label noise and with independent label noise. Results are averaged over 5 random seeds.

| Noise Rate 0% | MR | CR | MPQA | SUBJ | TREC |
|---|---|---|---|---|---|
| Fine-tuning | 83.37$\pm$0.70 | 83.29$\pm$0.80 | 87.56$\pm$0.70 | 93.14$\pm$0.42 | 83.28$\pm$0.86 |
| $\ell^2$-PGM | 84.16$\pm$0.41 | 83.87$\pm$0.66 | 87.77$\pm$0.62 | 93.16$\pm$0.21 | **84.48$\pm$0.52** |
| REGSL (ours) | **84.20$\pm$0.47** | **84.35$\pm$0.60** | **87.95$\pm$0.65** | **93.50$\pm$0.17** | 83.73$\pm$0.75 |
| Noise Rate 40% | MR | CR | MPQA | SUBJ | TREC |
| Fine-tuning | 82.91$\pm$0.25 | 77.67$\pm$1.11 | 82.85$\pm$0.98 | 90.78$\pm$0.88 | 70.80$\pm$0.22 |
| $\ell^2$-PGM | 83.45$\pm$0.28 | 79.47$\pm$0.37 | 84.03$\pm$0.41 | 72.14$\pm$0.21 | 73.48$\pm$0.90 |
| REGSL (ours) | **83.54$\pm$0.40** | **79.89$\pm$0.51** | **84.15$\pm$0.99** | **72.48$\pm$0.45** | **74.80$\pm$0.87** |

Table 8: Comparing $\sum_{i=1}^{L} D_i^2$ between layer-wise and constant regularization.

| | Aircrafts | CUB-200-2011 | Caltech-256 | Stanford-Cars | Stanford-Dogs | Flowers | MIT-Indoor |
|---|---|---|---|---|---|---|---|
| Layer-wise | 250.20 | 840.09 | 90.42 | 4150.02 | 20.72 | 252.45 | 57.16 |
| Constant | 2465.05 | 1225.69 | 195.72 | 2480.08 | 211.78 | 3151.28 | 295.49 |

start step $E_c$ is searched in $\{5, 8, 10, 13, 15\}$. Label correction threshold $p$ is set as $0.90$ in all experiments. The validation set sizes used for evaluating these hyperparameters choices range from 536 to 5120 (cf. Table 1). For the results in Table 2, we search for 20 different trials of the hyperparameters for the proposed algorithm and the baselines. For the results in Table 3, we search 20 times on the self-labeling parameters and 20 times on the regularization parameters. For the few-show learning experiments in Table 4, we use the validation set of Mini-ImageNet, following the training procedure of Dhillon et al. (2019) and search for 20 different trials.

We report the results for baselines by running the official open-sourced implementations. We describe the hyperparameters space for baselines as follows. For GCE (Zhang and Sabuncu, 2018), we search the factor $q$ in their proposed truncated loss $\mathcal{L}_q$ in $\{0.4, 0.8, 1.0\}$ and set the factor $k$ as 0.5. We also search the start pruning epoch in $\{3, 5, 8, 10, 13\}$. For SCE (Wang et al., 2019), we search the $\alpha$ and $A$ in their proposed symmetric cross entropy loss. The factor $\alpha$ is searched in $\{0.01, 0.10, 0.50, 1.00\}$, and the factor $A$ is searched in $\{-2, -4, -6, -8\}$. For DAC (Thulasidasan et al., 2019), we search the start abstention epoch in $\{3, 5, 8, 10, 15\}$ and set the other hyperparameters as the same in their paper. For APL (Ma et al., 2020), we choose the active loss as normalized cross entropy loss and the passive loss as reversed cross entropy loss. We search the loss factor $\alpha$ in $\{1, 10, 100\}$ and $\beta$ in $\{0.1, 1.0, 10\}$. For ELR (Liu et al., 2020), we search the momumtum factor $\beta$ in $\{0.5, 0.7, 0.9, 0.99\}$ and the weight factor $\lambda$ in $\{0.05, 0.3, 0.5, 0.7\}$. For SAT (Huang et al., 2020), the start epoch is searched in $\{3, 5, 8, 10, 13\}$, and the momumtum is searched in $\{0.6, 0.8, 0.9, 0.99\}$. We use the same number of trials in tuning hyperparameters for baselines.

## B.2 Extended experimental results

We present additional results. First, we apply our algorithm to another data set with label noise (same label noise as Table 3). Second, we apply our algorithm to sentence classification tasks.

**Additional results under label noise.** We report the test accuracy results of our algorithm on CUB-200-2011 and Flowers data sets with independent label noise in Table 5. From the table, we show that our algorithm still outperforms previous methods by a significant margin, which aligns with our results of the MIT-Indoor data set in Table 3.

**Results on ChestX-ray14.** We apply our approach to medical image classification, which is a more distant transfer task (relative to the source data set ImageNet). We report the mean AUROC (averaged over predicting all 14 labels) on the ChestX-ray14 data set in Table 6a. With layer-wise regularization, we see **0.39**% improvement compared to the baseline methods.

**Results on ViT.** We apply our approach to fine-tuning pre-trained ViT (Dosovitskiy et al., 2020) from noisy labels, under the same setting as Table 3). Table 6b shows the result. First, we find that our approach improves upon the best regularization methods by **1.17**% averaged over two settings. Second, we find that our approach also improves upon self-labeling by **13.57**% averaged over two

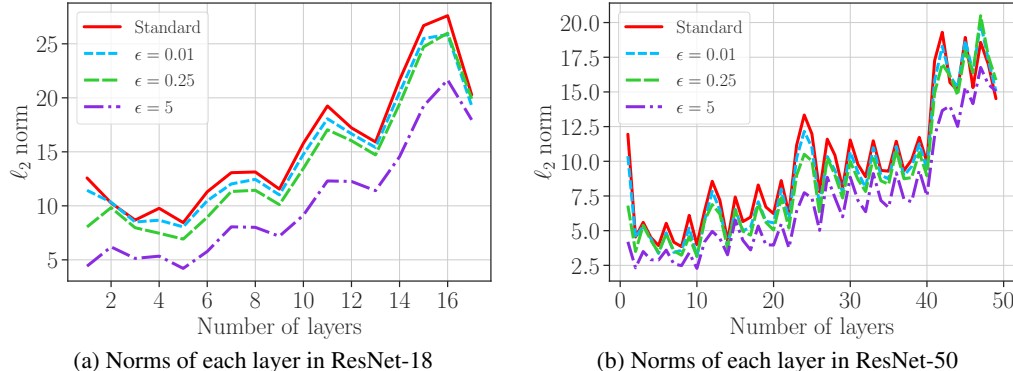

(a) Norms of each layer in ResNet-18        (b) Norms of each layer in ResNet-50

Figure 6: Frobenious norms of each layer in ResNet-18 pre-trained with standard and adversarial training methods from different noise levels ($\epsilon = 0.01, 0.25, 5$). Higher $\epsilon$ implies lower norms.

settings. These two results again highlight that both regularization and self-labeling contribute to the final result. While regularization prevents the model from over-fitting to the random labels, self-labeling injects the belief of the fine-tuned model into the noisy data set.

**Extension to text classification.** We apply our algorithm in text data domains. We conduct an experiment on sentiment classification tasks using a three-layer feedforward neural network (or multi-layer perceptron). We considered six text classification data sets: SST (Socher et al., 2013), MR (Pang and Lee, 2005), CR (Hu and Liu, 2004), MPQA (Wiebe et al., 2005), SUBJ (Pang and Lee, 2004), and TREC (Li and Roth, 2002). We use SST as the source task and one of the other tasks as the target task. We compared our proposed algorithm with fine-tuning and L2-PGM (Gouk et al., 2021). In Table 7, we report the results evaluated under a setting with no label noise and a setting with independent label noise (same setting in Table 3). The results show our proposed algorithm can outperform these baseline methods under both settings.

### B.3 Ablation studies

We describe several ablation studies. First, we compare the performance between using the Frobenius norm and the MARS norm proposed in Gouk et al. (2021). Second, we show the norm values of the adversarial robust pre-trained models in each layer and explain their improvement for fine-tuning on downstream tasks. Third, we compare our proposed layer-wise constraints and a constant and discuss their correlation to the generalization performance in Theorem 4.1. Fourth, we analyze the role of regularization in our proposed self label-correction method. Finally, we conduct an ablation study to study the influence of three components in our algorithm.

**Comparing layer-wise and constant regularization:** $\sum_{i=1}^{L} D_i^2$**.** To show the bound in Proposition 4.1 is nonvacuous, We compare the value of $\sum_{i=1}^{L} D_i^2$ from Equation 4 under both layer-wise constraints and constant regularization in Table 8. The values for every $D_i$ are from the parameters set in Table 2 for these data sets. We can see that layer-wise constraints incur a smaller value of $\sum_{i=1}^{L} D_i^2$ compared to constant regularization, which correlates with their better generalization performance.

**Comparing norms of adversarial robust pre-trained models.** We empirically observe that adversarially robust pre-trained models have lower Frobenius norms over each layer shown in Figure 6. The lower norms of robust models would induce smaller generation error, which explains their improvement in fine-tuning downstream tasks.

**Comparing label-correction precision with and without regularization.** We observe from Figure 5a that combining self-labeling and regularization is more effective than using just self-labeling. This is evidenced by the gap between correction with regularization and without regularization in the figure. In Table 9, we show the denominator of relabeling precision (i.e., the overall number of data points that are relabeled in line 7 of Algorithm 1) in the correction process. We find that the denominator is different between with and without regularization. The denominator is significantly smaller with regularization than without regularization, indicating that regularization prevents the model from over-fitting the noisy labels. The denominator decreases only with regularization. Without

Table 9: Denominator (total number of relabeled data points) and numerator (the number of correctly relabeled data) for calculating precision of label-correction in Figure 5a.

| Number of epoch | 1 | 4 | 7 | 10 | 13 | 16 | 19 | 21 |
|---|---|---|---|---|---|---|---|---|
| denominator w/ regularization | 28 | 32 | 32 | 29 | 26 | 24 | 21 | 19 |
| numerator w/ regularization | 15 | 17 | 16 | 13 | 12 | 10 | 9 | 8 |
| denominator w/o regularization | 56 | 66 | 64 | 64 | 62 | 58 | 59 | 56 |
| numerator w/o regularization | 34 | 33 | 28 | 26 | 21 | 18 | 17 | 14 |

Table 10: Top-1 test accuracy of using different norms for fine-tuning ResNet-101 pre-trained on the ILSVRC-2012 subset of ImageNet. Results are averaged over 3 random seeds.

|  | Aircrafts | CUB-200-2011 | Caltech-256 | Stanford-Cars | Stanford-Dogs | Flowers | MIT-Indoor |
|---|---|---|---|---|---|---|---|
| PGM (MARS) | 74.34±0.15 | 81.14±0.16 | 83.28±0.31 | 89.01±0.18 | 87.90±0.10 | 93.45±0.23 | 78.48±0.29 |
| PGM ($\ell^2$) | 74.90±0.26 | 81.23±0.32 | 83.25±0.33 | 88.92±0.39 | 86.48±0.28 | 93.23±0.34 | 77.31±0.30 |
| Ours (MARS) | 75.10±0.32 | 82.03±0.32 | 85.33±0.33 | 89.29±0.25 | 90.94±0.28 | 93.67±0.34 | 79.40±0.30 |
| Ours ($\ell^2$) | 75.32±0.23 | 82.24±0.21 | 84.90±0.16 | 89.14±0.22 | 89.58±0.13 | 93.82±0.35 | 79.30±0.31 |

Table 11: Removing any component in our algorithm leads to worst performance. Results are on the Indoor data set with independent label noise. Results are averaged over 3 random seeds.

| Indoor | independent noise | | | | correlated noise |
|---|---|---|---|---|---|
|  | 20% | 40% | 60% | 80% | 25.18% |
| REGSL (ours) | **72.51±0.46** | **68.13±0.16** | **57.59±0.55** | **34.08±0.79** | **70.12±0.83** |
| w/o regularization | 71.94±0.43 | 67.84±0.38 | 57.24±0.43 | 33.78±0.30 | 69.43±0.36 |
| w/o label correction | 70.92±0.41 | 59.10±0.24 | 47.81±0.35 | 28.42±0.46 | 69.78±0.34 |
| w/o label removal | 70.32±0.65 | 66.57±0.76 | 55.37±0.28 | 29.43±0.88 | 67.96±0.49 |
| w/o self-labeling | 70.23±0.25 | 64.40±0.58 | 54.20±0.68 | 32.54±0.43 | 69.05±0.09 |

regularization, the denominator remains roughly the same. Additionally, the number of incorrectly labeled data points by the self-labeling is much higher (e.g., 42 vs. 11 at epoch 21).

**Comparing $\ell_2$ norm and the MARS norm.** We report the results of using different norms for constraints in our algorithm in Table 10. We compare the performance of the Frobenius norm ($\ell^2$), and the MARS norm proposed in (Gouk et al., 2021). From the table, we show that the results of the two norms are similar. In the paper, we focus on the Frobenius norm for our discussion.

**Influence of different components in** REGSL. We study the influence of each component of our algorithm: layer-wise constraint, label correction, and label removal. We remove these components, respectively, and run the same experiments on the MIT-Indoor dataset with different kinds of label noise. Furthermore, we also include a row of results using only layer-wise constraints without self-labeling (containing label correction and removal). As shown in Table 11, removing any component from our algorithm could hurt the performance. This suggests that only incorporating these components can prevent both over-fitting and label memorization of models. In addition, We can see from Table 11 that when the noise rate is 20%, the self-labeling part (including label correction and removal) is more critical than regularization. When the noise rate is 40% or higher, the label correction part is the most important.

## C  Limitation

Our work focuses on the regularization and robustness properties of fine-tuning. We mainly focus on feedforward neural networks, including convolutional neural networks. Thus, we have not considered Other settings such as recurrent neural networks in this work. We mainly focus on independent label noise for evaluating the robustness. Other types of label noise, for example, using weak supervision, are not well-understood. This is left for future work. Finally, our treatment of label noise using self-labeling is based on empirical heuristics. It would be fascinating to come up with a more principled approach to tackle label noise.