# OpenReview forum: "Improved Regularization and Robustness for Fine-tuning in Neural Networks"
_NeurIPS.cc/2021/Conference — NeurIPS 2021 Poster_

### Official Review · Reviewer_a4M8 · 2021-07-15

**Rating:** 6
**Confidence:** 3

**Summary:**

The paper analyses the fine-tuning of neural networks. It provides a PAC-Bayes bound for the generalization of a fine-tuned network. The paper then provides three tricks to improve finetuning: layerwise regularization, label correction, and label removal. The resulting fine-tuning algorithm improves performance on various image classification tasks, including those with artificially-added label noise.

**Limitations And Societal Impact:**

Limitations and impact are adequately acknowledged.

**Main Review:**

*Summary*

Overall the paper is well written. The theoretical analysis provides some interesting motivation. I am not an expert on PAC-Bounds, so I cannot comment whether the result is of significant interest to the theoretical community, nor can I comment on whether the bound is tight or even non-vacuous. The resulting algorithm is intuitive and does seem to offer some improvements on real world datasets.

However, the main weaknesses for me are
* The claim "outperforms state-of-the-art" appears far too strong.
* The connection between the theoretical analysis and the algorithm appears quite weak.
* There is not sufficient detail on the theoretical analysis for someone unfamiliar with PAC-bounds to understand the implications.

Details below

*Originality*

The heuristic algorithmic improvements seem nice, offering modest improvements in performance. However, these are relatively small extensions of existing approaches.

*Quality*

Overall the contribution both of a theoretical result, and solid empirical numbers is convincing. However, I have a few concerns

1) The claim of "state-of-the-art on seven transfer tasks" seems to be hugely exaggerated, taking two examples

CUB-200-2011: the paper reports 82.24%, while [1] reports 91.7%, and many algorithms have reported over 90%.
Flowers: the paper reports 93.82%, while [2] reports 99.74%, and many algorithms have reported over 99%.

Perhaps the authors mean "state-of-the-art if limited to a particular ResNet101 pre-trained on ImageNet", however, "baseline" would then be more appropriate.

I do not believe that the method needs to attain state-of-the-art to demonstrate its validity, but the claim is too strong, and the fact that the scores are so far behind state-of-the-art (fine-tuning a stronger public pre-trained network is not expensive) brings into doubt the validity of the conclusions.

2) The connection between the PAC-Bayes bound and the algorithm appears weakly motivated. For example, the bound indicates that larger distances moved indicate a lower generalization error (which is intuitive given the Gaussian prior/posterior and KL[P|Q] term). However, the motivation for layerwise regularization (as opposed to the existing methods of constant regularization) is empirical. The bound does not appear to speak to layerwise differences.

3) The algorithm introduces many hyperparameters. While the hyparameter sweeps are listed in the Appendix, it was not clear exactly how many hyperparameters are swept for each algorithm, including the baselines. Could the authors provide a concise summary of the size of the search space for each method?

*Clarity*

The paper is generally well written. One part that is hard to follow is Section 4.1. For example, the "noise stability l(...)" does not appear in Eqn (4), so it is not clear how these are connected. This section could be expanded and more details provided.

A few typos spotted
* genrealization
* disturbation
* over-fitiing
* regualrized
* line 227: max(p) <= t,  should this be  >= ?

*Significance*

The set of tricks to improve fine-tuning could be useful to practitioners. However, it does not constitute a very significant algorithmic advance.

Without further context, the theoretical result does not appear significant (is the bound tight/trivial?), however, as mentioned above I am not an expert on theory, so I will leave it to others to assess the significance of this result.


[1] [TransFG: A Transformer Architecture for Fine-grained Recognition](https://arxiv.org/abs/2103.07976)

[2] [An Image is Worth 16x16 Words: Transformers for Image Recognition at Scale](https://arxiv.org/abs/2010.11929v2)



POST REBUTTAL

The response provided additional context and experiments, and the claims of SOTA will be tempered. Increasing my score.


**Time Spent Reviewing:**

5

---

> ### Author Response · Authors · 2021-08-10
> **Response**
>
> We thank the reviewer for the constructive feedback, which helps improve our work. We provide detailed discussions to the reviewer's comments below.
>
> **>>> “The claim of "state-of-the-art on seven transfer tasks" seems to be hugely exaggerated.”**
>
> We thank the reviewer for pointing out this issue and apologize for any confusion. We will change the phrasing of “achieving state-of-the-art” in lines 71 and 251 to “outperforming baseline methods” (the latter is used in the abstract, cf. line 18).
>
> As the reviewer pointed out, recent architectural innovations in computer vision can result in more accurate predictions on the CUB and flowers datasets. Since ResNet101 pre-trained on ImageNet is a widely used model, and over-fitting is a prevalent issue in fine-tuning, we think our results are valuable in their own right.  Additionally, our analysis offers a pipeline that can potentially apply to different settings (e.g., we have conducted exploratory analysis on text classification tasks in our response to another reviewer, and obtained encouraging results):
>
> - Run (vanilla) fine-tuning on the pre-trained initialization.
> - Plot the distance “traveled” similar to Figure 2.
> - Encode the layer-wise distance patterns using explicit regularization constraints.
>
> We will follow the reviewer’s suggestion and change our phrasing in the next version of the paper.
>
> **>>> “The connection between the PAC-Bayes bound and the algorithm appears weakly motivated. However, the motivation for layerwise regularization (as opposed to the existing methods of constant regularization) is empirical. The bound does not appear to speak to layerwise differences.”**
>
> We thank the reviewer for the intriguing comment. We agree with the reviewer that the motivation for layer-wise constraints comes from the empirical observation in Figure 2. To see why the bound is helpful, we compare the value of $\sum_{i=1}^L D_i^2$ (from equation (4)) under both layer-wise constraints and constant regularization:
>
> |            | CUB-200-2011 | Flowers |
> | :--------: | :----------: | :-----: |
> | Layer-wise |    840.09    | 252.45  |
> |  Constant  |   1225.69    | 3151.28 |
>
> The values for every $D_i$ are from the parameters set in Table 2 for these datasets. We can see that layer-wise constraints incur a smaller value of $\sum_{i=1}^L D_i^2$ compared to constant regularization, which correlates with their better generalization performance.
>
> **>>> “The algorithm introduces many hyperparameters. While the hyper-parameter sweeps are listed in the Appendix, it was not clear exactly how many hyperparameters are swept for each algorithm, including the baselines. Could the authors provide a concise summary of the size of the search space for each method?”**
>
> Thanks for asking this question. For the results in table 2, we searched for 20 different trials of the hyperparameters for the proposed algorithm and the baselines. For the results in table 3, we searched 20 times on the self-labeling parameters and 20 times on the regularization parameters. The search space is
>
> - (Base) distance value $D$: [0.05, 10].
> - Distance scaling factor $\gamma$: [1, 5].
> - Self-labeling parameters (four of them): 100 combined.
>
> The search cost for the baselines is the same. We will include these discussions in the main text in the updated paper.
>
> **>>> “The set of tricks to improve fine-tuning could be useful to practitioners. However, it does not constitute a very significant algorithmic advance. Without further context, the theoretical result does not appear significant (is the bound tight/trivial?).”**
>
> We appreciate the reviewer’s comment on this work. We partially agree that some of the main contributions leverage existing tools such as distance-based regularization and the PAC-Bayesian bounds. We explain the significance of this work in three aspects.
>
> - First, regularization is vital for preventing fine-tuning from overfitting, as highlighted by the work of Gouk et al., (2020) and references therein. Recent work (e.g., Jiang et al., 2019) has found PAC-Bayesian bounds highly correlated with empirical performance in neural networks. However, applying these bounds to understand fine-tuning is relatively unexplored.
>
> - Second, the conceptual contribution of the theoretical result is introducing the PAC-Bayesian analysis to fine-tuning. The best bound that we can obtain (by tuning $\sigma$ and invoking equation (4) in McAllester (2013)) scales as $$\text{O}\left(\sqrt{\frac{\Big(\prod_{i=1}^L (B_i + D_i)\Big) \cdot \Big(\sum_{i=1}^L \frac{1}{B_i + D_i}\Big) \cdot \Big(\sum_{i=1}^L D_i^2\Big)}{n^{(t)}}}\right).$$
>   - Compared to Theorem 2 of Gouk et al. (2021), the above result eliminated the dependence on the width of each layer (i.e., the product of $n_k$), while incurring an extra factor that is equal to $\sum_{i=1}^L D_i^2$. The value of this factor (cf. the table above) is much smaller than the product of layer widths $n_k$ for the ResNet-101.
>
> - Finally, the above analysis has led to several practical insights. One new insight of this paper (using noise stability) is that fine-tuning from a pre-trained initialization results in a more stable model (in the sense of loss after random Gaussian perturbations) compared to fine-tuning from a random initialization (cf. table 2(b)). Another new insight is the interaction between regularization and self-labeling under label noise (cf. figures 3 and 4).
>
> We hope the above discussion helps. Let us know if we have alleviated your concern on the limited significance of this work. We appreciate your comments!
>
> **References**
>
> - Jiang, Y., Neyshabur, B., Mobahi, H., Krishnan, D., & Bengio, S. (2019). Fantastic generalization measures and where to find them. *arXiv preprint arXiv:1912.02178*.

---

> > ### Comment · Reviewer_a4M8 · 2021-08-18
> > **Acknowledged**
> >
> > Details of the hyperparameter search, and tempering the claims of "state of the art" appreciated. I am still not fully convinced by the connection between the theoretical analysis and proposed algorithmic improvements. But I will consider adjusting my score (up) following the discussion.

---

> > > ### Author Response · Authors · 2021-08-23
> > > **Thanks for the response**
> > >
> > > Thanks for the comment (we have been working on the response). We appreciate the reviewer’s time for looking into our response and are glad to find that some of the clarifications helped. Below we provide further discussion on the follow-up comment.
> > >
> > > **>>> "Background on PAC-Bounds and implications of these bounds for fine-tuning"**
> > >
> > > In light of the reviewer’s earlier comment on “a lack of detail on the theoretical analysis for someone unfamiliar with PAC-bounds to understand the implications,” we thought it might be helpful to begin with a general background on the PAC-Bounds. Recent works (e.g., Bartlett et al., 2017 and Neyshabur et al., 2017) introduced generalization bounds for multi-layer neural networks in an attempt to explain why neural network models generalize well despite having more trainable parameters than the number of training examples. On the one hand, the VC dimension of a neural network is known to be roughly equal to its number of parameters (Bartlett et al., 2019). On the other hand, the number of parameters is not a good capacity measure for neural nets, as evidenced by the popular work of Zhang et al., 2021. The bounds of Bartlett et al., 2017 (and follow-up works) provide a more meaningful notion of “capacity” compared to VC-dimension.
> > >
> > > While these bounds constitute an improvement compared to classical learning theory, it is unclear if these bounds are tight/non-vacuous (as the reviewer rightly points out). One response to this criticism is a computational framework from Dziugaite and Roy, 2017. That work shows that directly optimizing the PAC-Bayes bound can get a much smaller bound and low test error simultaneously (see also Zhou et al., 2018 for a large scale study). The recent work of Jiang et al., 2019 further compared different complexity notions and noted that the ones given by PAC-Bounds correlate the most with empirical performance.
> > >
> > > The PAC-bounds we showed for fine-tuning are based on the analysis in Neyshabur et al., 2017. As discussed in the initial response, the bounds (in particular, the part on $\sum_i D_i^2$) gives an intuitive way to think about how one should design these distance constraints. While a larger value of $D_i$ increases the network’s “capacity,” the generalization bound gets worse as a result. Thus, there is a tradeoff in setting $D_i$. The heuristic improvement we showed may be viewed as a fast and easy-to-use rule of thumb for setting these constraints in practice. This is inspired by (if not directly connected to) the analysis; in turn, the bound is useful for explaining our experiments compared to constant regularization (as discussed in the previous response).
> > >
> > > Another implication from the analysis is that pre-trained models are less susceptible to noise when fine-tuned (and perform better) than randomly initialized models. This finding reaffirms the line of results discussed above in the context of fine-tuning. This also partly explains why adversarially robust pre-trained models can sometimes outperform standard pre-trained models for fine-tuning (as discussed in our response to another reviewer).
> > >
> > > We hope this discussion helps and plan to include them in the appendix in the updated paper. Let us know if this discussion alleviates the concern around the lack of background and connection between the analysis and the algorithm. We thank the reviewer again for surfacing these issues, which we were previously unaware of!
> > >
> > > **>>> "Issues regarding clarity and typos"**
> > >
> > > Just realized we have not yet acknowledged these issues in the previous response. We will correct them in the updated paper.
> > >
> > > **>>> "Extension to Vision transformers (ViT)"**
> > >
> > > We would like to follow up on the reviewer’s comment on fine-tuning using pre-trained ViT (Dosovitskiy et al., 2020). With some more time, we conducted a preliminary study using our method and found out that regularization and self-labeling are still helpful for fine-tuning ViT given noisy labels.
> > >
> > > First, we were able to replicate the reviewer’s comment that fine-tuning pre-trained ViTs leads to more accurate predictions (e.g., over 99% test accuracy on the Flowers dataset). This only requires early stopping, which may be viewed as a form of implicit regularization.
> > >
> > > Second, we further studied our method for fine-tuning the pre-trained ViT given noisy labels (i.e., the same setting as Table 3) and found some interesting results. We used the ViT-Base model pre-trained on the ImageNet-21k dataset (Dosovitskiy et al., 2020). We followed the same fine-tuning procedure described in Dosovitskiy et al. (2020) and fine-tuned it for 30 epochs. The hyperparameters for our method are searched using the same process as in ResNet-101. The test accuracies for our method (and each component of it) are in the following table.
> > >
> > > | Methods | Independent noise, 80% |
> > > | :------ | :-------------------------: |
> > > | Fine-tuning     |        32.06 ± 0.17 |
> > > | Layer-wise reg. |    44.63 ± 0.36     |
> > > | Self-labeling   |   41.69 ± 0.16      |
> > > | Our method      |   55.05 ± 0.67      |
> > >
> > > This table shows that both the regularization and the self-labeling part contribute to the final result on the MIT-67 indoor scenes dataset (Sharif Razavian et al., 2014). We expect similar results to hold for other noise rates described in table 3. Our intuition behind these results is that while regularization prevents the fine-tuned model from overfitting to the random label, self-labeling injects the fine-tuned model’s belief into the noisy dataset.
> > >
> > > We thank the reviewer again for suggesting these two papers! We will add the two citations and include a table similar to Table 3 for fine-tuning ViT in the updated paper. Let us know if you have any questions about this experiment or other parts of the paper. We are happy to continue the discussion.
> > >
> > > **References**
> > >
> > > - Bartlett, Peter, Dylan J. Foster, and Matus Telgarsky. "Spectrally-normalized margin bounds for neural networks." arXiv preprint arXiv:1706.08498 (2017).
> > > - Neyshabur, Behnam, Srinadh Bhojanapalli, and Nathan Srebro. "A pac-bayesian approach to spectrally-normalized margin bounds for neural networks." arXiv preprint arXiv:1707.09564 (2017).
> > > - Bartlett, P. L., Harvey, N., Liaw, C., & Mehrabian, A. (2019). Nearly-tight VC-dimension and pseudodimension bounds for piecewise linear neural networks. The Journal of Machine Learning Research.
> > > - Zhang, C., Bengio, S., Hardt, M., Recht, B., & Vinyals, O. (2021). Understanding deep learning (still) requires rethinking generalization. Communications of the ACM, 64(3), 107-115.
> > > - Dziugaite, G. K., & Roy, D. M. (2017). Computing nonvacuous generalization bounds for deep (stochastic) neural networks with many more parameters than training data. arXiv preprint arXiv:1703.11008.
> > > - Zhou, W., Veitch, V., Austern, M., Adams, R. P., & Orbanz, P. (2018). Non-vacuous generalization bounds at the imagenet scale: a PAC-bayesian compression approach. arXiv preprint arXiv:1804.05862.
> > > - Dosovitskiy, A., Beyer, L., Kolesnikov, A., Weissenborn, D., Zhai, X., Unterthiner, T., ... & Houlsby, N. (2020). An image is worth 16x16 words: Transformers for image recognition at scale. arXiv preprint arXiv:2010.11929.

---

> > > > ### Comment · Reviewer_a4M8 · 2021-09-02
> > > > **Acknowledged2**
> > > >
> > > > Thanks for the additional explanation explaining how the provenance of the bound and the connection the observed results.  The new experiments I think strengthen the experimental side further. I will increase by score.

---

### Official Review · Reviewer_uNxM · 2021-07-16

**Rating:** 6
**Confidence:** 3

**Summary:**

This paper studies the aspects of fine-tuning a model (pre-trained on a separate dataset) in order to learn a new task. It has the following contribution:

(i) Generalization properties of fine-tuning explained by the PAC-Bayes generalization bound that depends on two terms: (a) distance between each layer from the pre-trained model and (b) noise stability of the pre-trained model.

(ii) Inspired by the generalization analysis, the paper presents an algorithm to perform fine-tuning. This algorithm includes (a) regularization term that computes the distance between the current parameters and the pre-trained model, and (b) iteratively corrects mislabeled examples where model has high confidence and down-weighting less confident examples.

(iii) Proposed algorithm is evaluated on a suite of benchmark tasks including transfer learning and few shot classification tasks.



**Limitations And Societal Impact:**

Yes.

**Main Review:**

Strengths:
---------------

- PAC-Bayes generalization bound for fine-tuning explains the generalization performance of the model obtained from fine-tuning. In hindsight, the terms appearining in the bounds look intuitive (i.e. distance from the pre-trained model and the noise stability of the model). It also explains why models trained with adversarial noise result in better fine-tuned performance.

Weakness:
---------------

- PAC-Bayes bound shows two terms. First one is the distance from the pre-trained model, which the proposed algorithm incorporates as it is. But the second term that is the noise stability of the model, is handled in a heuristic way. Instead a principled approach should have been to incorporate the noise stability by adding some sort of penalty term (similar to the prior used in the PAC-Bayes analysis or some sort of adversarial noise).

- Heuristic to incorporate the label correctness seems to have some issues. (see below)

Questions for Authors:
---------------

- Label correction is done by thresholding the confidence of the classifier, in this particular case the class scores less than  a threshold. Did you try any other heuristics for label correction like entropy, etc.?

- What is the rational in down-weighting low confident examples? Why is it assumed that the network cannot learn these examples at all?

- Similarly, iteratively correcting the mislabeled examples does not seem like a good idea. Did the authors verify qualitatively how many times these "corrections" were indeed correct?


Writing Clarity:
---------------

Paper is clearly written and easy to understand.



**Time Spent Reviewing:**

7

---

> ### Author Response · Authors · 2021-08-10
> **Response**
>
> Thanks for the detailed review, which helps improve our work. We provide detailed discussions to the reviewer's comments below.
>
> **>>> “The second term of the generalization bound that is the noise stability of the model, is handled in a heuristic way. Instead a principled approach should have been to incorporate the noise stability by adding some sort of penalty term.”**
>
> We thank the reviewer for this excellent suggestion. We agree that incorporating the noise stability by adding some penalty terms is an interesting question. Currently, our algorithm only involves layer-wise constraints. For example, one promising direction is to extend the layer-wise constraints with some form of penalty on the Hessian (e.g., Foret et al., 2021). While we cannot provide a conclusive statement because of the short rebuttal period, we are interested in exploring this direction in future work. We will add these discussions to the updated paper.
>
> **>>> “Label correction is done by thresholding the confidence of the classifier, in this particular case the class scores less than a threshold. Did you try any other heuristics for label correction like entropy, etc.?”**
>
> We thank the reviewer for the suggestion. We have not yet tried using the entropy for thresholding. Our intuition is that the examples with low confidence and high entropy may overlap. Thus, our intuition is that the results using entropy will likely be similar.
>
> **>>> “What is the rationale for down-weighting low confident examples? Why is it assumed that the network cannot learn these examples at all?”**
>
> Thanks for these questions. Our rationale is that these low-confidence examples likely have the incorrect label. Thus, down-weighting them reduces the fraction of noisy datapoints in the training set (and the gradient calculations). Figure 4(b) verifies the hypothesis since the samples with true labels have much higher weights than the samples with incorrect labels.
>
> We cannot assume the network will “learn” all of these examples because the sample sizes from the training set are relatively small (cf. Table 1). We will include these clarifications in Section 4.2 in the updated paper.
>
> **>>> “Did the authors verify qualitatively how many times these "corrections" were indeed correct**?”
>
> Yes, the label correction accuracy (for the MIT-Indoor dataset) can be found in Figure 4(a) in Section 4.2. This figure shows that the accuracy of the label correction part is highest in the beginning. Additionally, the layer-wise constraints can enhance the accuracy of these corrections.
>
> **References**
>
> - Foret, P., Kleiner, A., Mobahi, H., & Neyshabur, B. (2020). Sharpness-aware minimization for efficiently improving generalization. *arXiv preprint arXiv:2010.01412*.

---

> > ### Comment · Reviewer_uNxM · 2021-08-31
> > **Response to Author Rebuttal**
> >
> > Thank you for the response.
> >
> > I agree with the observation that low confidence and high entropy regions would most likely overlap.
> >
> > I have also looked at some of the updated empirical results that you have posted during this period. I really appreciate the quick turn around with various experiments on the new datasets. These new results do help in understanding the significance of this work.
> >
> > I do think the noise stability term introduced in the generalization bound should be incorporated in future work. It would help in avoiding the heuristics related to the noise part.
> >
> > **Seeking Clarification**
> >
> > Thank you pointing out the ablative experiment with "corrections" evaluation. Could you elaborate a bit on why do you define "correction accuracy" as the ratio of the number of corrected labels to the number of changed labels in each epoch? I understand what is the number of corrected labels, but what exactly is the number of changed labels in an epoch? Does it corresponds to the label changes during training? If so, shouldn't the denominator also as you go down towards the end of the training phase? In which case, your conclusion is not immediately clear from this figure 4(a).

---

> > > ### Author Response · Authors · 2021-08-31
> > > **Response to the clarification questions**
> > >
> > > Thanks for the kind response. We think incorporating noise stability in a principled way is an interesting question for future work. We will include a discussion regarding this point in the updated paper. Below we respond to the clarification questions.
> > >
> > > **>>> "Could you elaborate a bit on why do you define "correction accuracy" as the ratio of the number of corrected labels to the number of changed labels in each epoch? I understand what is the number of corrected labels, but what exactly is the number of changed labels in an epoch? Does it corresponds to the label changes during training?"**
> > >
> > > Yes, the number of changed labels in each epoch is the number of data points whose labels are flipped due to applying line 7 of Algorithm 1 during that epoch. For example, suppose that the labels of 20 data points are flipped in line 7, ten of which flips to that data point's correct label, then the correction accuracy is 50%. Thus, the correction accuracy indeed measures the precision of step 7 of Algorithm 1. We will clarify this description in the updated paper.
> > >
> > > **>>> "If so, shouldn't the denominator also as you go down towards the end of the training phase? In which case, your conclusion is not immediately clear from this figure 4(a)."**
> > >
> > > We clarify that the conclusion we would like to draw from this figure is that combining self-labeling and regularization is more effective than using just self-labeling. This is evidenced by the gap between the two lines in this figure.
> > >
> > > As the reviewer suggests, the denominator (i.e., the overall number of datapoints whose labels are changed in line 7 of Algorithm 1) is indeed different between with and without regularization (see results below).
> > > - The denominator is significantly smaller with regularization than without regularization, indicating that regularization prevents the model from overfitting to the noisy labels.
> > > - The denominator decreases only with regularization. Without regularization, the denominator remains roughly the same. Additionally, the number of incorrectly labeled data points by the self-labeling is much higher (e.g., 42 vs. 11 at epoch 21).
> > >
> > >
> > > | **epoch number**                       | 1    | 4    | 7    | 10   | 13   | 16   | 19   | 21   |
> > > | ---------------------------------- | ---- | ---- | ---- | ---- | ---- | ---- | ---- | ---- |
> > > | **denominator w/ regularization**  | 28   | 32   | 32   | 29   | 26   | 24   | 21   | 19   |
> > > | **numerator w/ regularization**    | 15   | 17   | 16   | 13   | 12   | 10   | 9    | 8    |
> > > | **denominator w/o regularization** | 56   | 66   | 64   | 64   | 62   | 58   | 59   | 56   |
> > > | **numerator w/o regularization**   | 34   | 33   | 28   | 26   | 21   | 18   | 17   | 14   |
> > >
> > > We hope these results help clarify our conclusion regarding the interaction between self-labeling and regularization.

---

### Official Review · Reviewer_YhjQ · 2021-07-18

**Rating:** 7
**Confidence:** 3

**Summary:**


The authors present a new analysis of PAC-Bayes generalization in the finetuning with regularization setting and analyze it’s properties in terms of layer wise L2 distances and noise robustness.  They propose a new modified algorithm for finetuning based on having different constraints on different layers as well as self-labeling.

**Limitations And Societal Impact:**

Societal impact and limitations (future work) are adequately covered

**Main Review:**

Originality/Clarity:
The paper is overall clearly presented. To the best of reviewers knowledge the work is original, including the bounds, analysis (e.g. motivating usefulness of robustness), and fine-tuning approach. The experiments are reasonably well done, using relevant baselines and datasets.

Significance:
Fine-tuning is still not fully understood from a theory point of view and this work contributes to improving our understanding. Furthermore, the proposed method shows promising results.

General Comments:
-Most of the tasks studied are closely related to the source task. It would be interesting to study the behavior of the proposed algorithm when more distant tasks are used (e.g. medical images).

-The connection to “explaining robustness” is a bit loose as the authors point out that adversarially robust models have lower operator norm. Is it possible to go a bit further than this and  illustrate this theoretically

-The authors illustrate how naturally the L2 distance for lower layers is smaller than higher layers, so why is there a need/advantage to incorporate this into the constraints if it already occurs? Related to that the authors motivate this phenomenon by referencing the standard interpretation of feature roles in lower and upper layers, I wonder if this is as well explained by the reduction in gradient norm with depth that typically occurs  in backprop.

-The ablations should include a baseline that shows the effect of layer-wise constraints without self-label



Overall, the paper has useful theoretical and empirical findings. Some further clarity and ablations can further improve it.

Post-rebuttal: The authors have reasonable addressed my primary concerns.


**Time Spent Reviewing:**

2

---

> ### Author Response · Authors · 2021-08-10
> **Response**
>
> We thank the reviewer for the constructive feedback, which helps improve our work. Below we provide detailed response to the comments.
>
> **>>> “It would be interesting to study the behavior of the proposed algorithm when more distant tasks are used (e.g. medical images).”**
>
> Thanks for the excellent suggestion. We think that our regularization method also applies to fine-tuning from more distant tasks. Our pipeline for identifying the distance constraints consists of three steps: (i) Run (vanilla) fine-tuning on the pre-trained initialization. (ii) Plot the distance “traveled” similar to Figure 2. (iii) Encode the layer-wise distance patterns using explicit regularization constraints.
>
> We conducted a preliminary experiment on the ChestX-ray14 dataset (Wang et al., 2017; Rajpurkar et al., 2017), containing 112120 frontal-view chest X-ray images labeled with 14 different diseases. We used a ResNet-18 pre-trained on ImageNet as the initialization and followed the same training procedure of Rajpurkar et al. (2017). We report the mean AUROC (averaged over predicting all 14 labels) of fine-tuning, L2-PGM, and our proposed algorithm:
>
> |   Methods   | AUROC |
> | :--------- | :--------: |
> | Fine-tuning |   0.8159   |
> |   L2-PGM    |   0.8235   |
> |    Our Method     |   **0.8274**   |
>
> These initial results show that our proposed algorithm can still help when more distant tasks are used (as the reviewer suggested). We will include the final results in the experimental section of the updated paper.
>
> **>>> “The connection to “explaining robustness” is a bit loose… Is it possible to go a bit further than this and illustrate this theoretically?”**
>
> Thanks for the suggestion. Recall that our current result (cf. Figure 5 in Section C.2) finds that models fine-tuned from adversarially robust initializations have weights with lower operator norms. To strengthen the result, we have measured the noise stability of models fine-tuned from adversarially robust initializations on the CUB-200-2011 and MIT-Indoor dataset. The noise stability measured in this table is the loss of the fine-tuned model with additional perturbation---every weight parameter adds a Gaussian random variable with standard deviation stated in the first column. This table shows that models fine-tuned from adversarially robust pre-trained initializations often have lower losses than models fine-tuned from pre-trained initializations.
>
> | CUB-200-2011 | Noise | Randomly-initialized |  Pretrained   | Adversarially-pretrained |
> | ------------ | :---: | :------------------: | :-----------: | :----------------------: |
> |              | 1e-2  |      3.77±0.42       | **1.45±0.13** |        1.76±0.09         |
> |              | 1e-3  |      0.82±0.07       |   0.62±0.03   |      **0.54±0.03**       |
> |              | 1e-4  |      0.81±0.04       |   0.61±0.03   |      **0.61±0.01**       |
>
> | MIT-Indoor | Noise | Randomly-initialized | Pretrained | Adversarially-pretrained |
> | ---------- | ----- | -------------------- | ---------- | ------------------------ |
> |            | 1e-2  | 2.51±0.34            | 1.11±0.09  | **0.97±0.07**            |
> |            | 1e-3  | 0.49±0.09            | 0.36±0.05  | **0.32±0.04**            |
> |            | 1e-4  | 0.44±0.03            | 0.33±0.02  | **0.30±0.04**            |
>
> Regarding the reviewer’s question of illustrating this theoretically, we think this is an interesting research question. One possible direction is to apply Taylor’s expansion on the adversarial training objective and relate the gradient and the Hessian to some regularization effects. These effects can then be measured empirically, similar to our analysis above. We leave this for future work. We will include the new study in Table 2(b) and discussion in the conclusion section.
>
> **>>> “Why is there a need/advantage to incorporate the layer-wise constraint into the constraints if it already occurs? I wonder if this is as well explained by the reduction in gradient norm with depth that typically occurs in back-prop.”**
>
> We thank the reviewer for these insightful questions. We clarify the advantages of the layer-wise constraints in two parts.
>
> - Figure 2a is drawn using fine-tuning w/ early stopping. While early stopping provides an implicit regularization, the layer-wise constraint encodes the regularization explicitly.
>
> - In our experiments, we have observed that using the same distance value for every layer only restricts the fine-tuned distance of the top layers. Layer-wise constraints further limit the fine-tuned space of the bottom layers to achieve layer-wise regularization fully.
>
> We also agree with the reviewer’s intuition that the empirical observation in Figure 2a is likely caused by the reduction in gradient norm during backpropagation.
>
> **>>> “The ablations should include a baseline that shows the effect of layer-wise constraints without self-label.”**
>
> We thank the reviewer for suggesting this ablation study. We have now included using layer-wise constraints without self-labeling in table 7 in section C.2. The results are shown below. layer-wise regularization w/o self-labeling performs worse than the first row of table 7.
>
> | MIT-Indoor | Independent Noise |            |            |            | Correlated Noise |
> | ---------- | :---------------: | :--------: | :--------: | :--------: | :--------------: |
> |            |        20%        |    40%     |    60%     |    80%     |      25.18%      |
> |            |    70.23±0.25     | 64.40±0.58 | 54.20±0.68 | 32.54±0.43 |    69.05±0.09    |
>
> We will include these results in Table 7 in the updated paper.
>
> **References:**
>
> - Wang, X., Peng, Y., Lu, L., Lu, Z., Bagheri, M., & Summers, R. M. (2017). Chestx-ray8: Hospital-scale chest x-ray database and benchmarks on weakly-supervised classification and localization of common thorax diseases. The IEEE conference on computer vision and pattern recognition.
> - Rajpurkar, P., Irvin, J., Zhu, K., Yang, B., Mehta, H., Duan, T., ... & Ng, A. Y. (2017). Chexnet: Radiologist-level pneumonia detection on chest x-rays with deep learning. arXiv preprint arXiv:1711.05225.

---

### Official Review · Reviewer_FQxu · 2021-07-18

**Rating:** 6
**Confidence:** 4

**Summary:**

Inspired by the empirical findings on fine-tuned distance in each layer and noise stability of model, this paper introduces a fine-tuning algorithm that incorporates layer-wise distance constraint and self label correction and removal. To prove the effectiveness of their algorithm on regularization and robustness, the authors provide experimental results on fine-tuning on different image datasets and fine-tuning on noisy label respectively.

**Limitations And Societal Impact:**

Yes.

**Main Review:**

Strengths
1.	The claims in this work are well supported by the PAC-Bayes generalization bound and the experimental results. The authors also provide codes in the supplementary material.
2.	This paper is well written and easy to follow. Their algorithm is related to distance-based regularization and noisy labels problems. The relations are clearly illustrated.

Weaknesses
1.	Overall, the paper is highly related to the previous work (Gouk et al. 2020) in terms of the distance-based regularization techniques and PAC-Bayes generalization bound, which slightly degrades the novelty of this work.
2.	The algorithm is the combination of three components: distance regularization, label correction and label removal. It is unclear which part is most important. To my understanding, the distance regularization in this work is $l^2$-pgm (Gouk et al. 2020) + layer wise scale factor. From the ablation study in appendix c.2, the accuracy of distance regularization + label removal is worse than the accuracy of $l^2$-pgm for most setting of independent noise. Seems like by adding the scale factor and label removal, the result is even worse.
3.	The authors only provide experiments on image dataset with relatively small size. Is this algorithm more effective on the convolutional layers? It might be better to add experiments in other application domains that involves more non-convolutional layers. There are six hyper-parameters in the algorithm. It might be challenging to set them with limited labeled data, especially for the label correction threshold and re-weighting temperature factor.



**Time Spent Reviewing:**

4 hours.

---

> ### Author Response · Authors · 2021-08-10
> **Response**
>
> We thank the reviewer for the insightful review, which helps improve our paper. Below we respond to the reviewer's comments one by one.
>
> **>>> “The paper is highly related to the previous work (Gouk et al. 2020)... which degrades the novelty of this work.”**
>
> We appreciate the reviewer’s comment. We hope to highlight three aspects that distinguish this work from Gouk et al., 2020 (as the reviewer rightly points out).
>
> - First, we utilize the PAC-Bayesian bounds for analyzing fine-tuning, whereas the proof of Gouk et al., 2020 uses Rademacher complexity. As discussed in the related work section, recent works have found the PAC-Bayesian bounds better correlated with empirical evidence than Rademacher complexity (e.g., Jiang et al., 2019). Inspired by these works, one new insight from our PAC-Bayesian analysis is that fine-tuning from a pre-trained initialization results in a more stable model (under random Gaussian perturbations) than fine-tuning from random initializations (cf. Table 2b).
> - Second, while our empirical analysis focuses on ResNet-101, we have also identified a pipeline that can potentially apply to other settings.
>   - Run (vanilla) fine-tuning on the pre-trained initialization.
>   - Plot the distance “traveled” similar to Figure 2.
>   - Encode the layer-wise distance patterns using explicit regularization constraints.
> - Finally, we have studied the robustness of fine-tuning given noisy labels, which is well-studied in supervised learning (e.g., Natarajan et al., 2013), but little is known about fine-tuning.
>
> **>>> “It is unclear which part of the algorithm is most important. In appendix c.2, the accuracy of distance regularization + label removal is worse than the accuracy of l2-pgm for settings of independent noise.”**
>
> We thank the reviewer for asking this question. Since each part of the algorithm plays a different role, they all contribute to the final performance. To answer the reviewer’s question, we can see that (based on Table 7) when the noise rate is 20%, the self-labeling part (including label correction and removal) is more critical than regularization. When the noise rate is 40% or higher, the label correction part is the most important. We will incorporate this discussion into the updated paper.
>
> **>>> “It might be better to add experiments in other application domains that involve more non-convolutional layers.”**
>
> Thanks for the suggestion. Under the reviewer’s suggestion, we conducted a preliminary experiment on text classification tasks using a three-layer feedforward neural network (or multi-layer perceptron) and obtained encouraging results.
>
> Setup. We considered six text classification datasets: SST, MR, CR, MPQA, SUBJ, and TREC ([more details here](https://github.com/harvardnlp/sent-conv-torch/tree/master/data)). We use SST as the source task and one of the other tasks as the target task. We compared our proposed algorithm with fine-tuning and L2-PGM (Gouk et al., 2020). We report the results evaluated under a setting with no label noise and a setting with independent label noise:
>
> | Noise Rate 0% |       MR       |       CR       |      MPQA      |      SUBJ      |      TREC      |
> | :------------ | :------------: | :------------: | :------------: | :------------: | :------------: |
> | Fine-tuning   |   83.37±0.70   |   83.29±0.80   |   87.56±0.70   |   93.14±0.42   |   83.28±0.86   |
> | L2-PGM        |   84.16±0.41   |   83.87±0.66   |   87.77±0.62   |   93.16±0.21   | **84.48±0.52** |
> | Ours          | **84.20±0.47** | **84.35±0.60** | **87.95±0.65** | **93.50±0.17** |   83.73±0.75   |
>
> | Noise Rate 40% | MR             | CR             | MPQA           | SUBJ           | TREC           |
> | -------------- | -------------- | -------------- | -------------- | -------------- | -------------- |
> | Fine-tuning    | 82.91±0.25     | 77.67±1.11     | 82.85±0.98     | 90.78±0.88     | 70.80±0.22     |
> | L2-PGM         | 83.45±0.28     | 79.47±0.37     | 84.03±0.41     | 72.14±0.21     | 73.48±0.90     |
> | Ours           | **83.54±0.40** | **79.89±0.51** | **84.15±0.99** | **72.48±0.45** | **74.80±0.87** |
>
> These are exploratory results, but overall, our proposed algorithm outperforms both baseline methods under both settings. We will include the final results in the experimental section of the updated paper.
>
> **>>> “There are six hyper-parameters in the algorithm. It might be challenging to set them with limited labeled data, especially for the label correction threshold and re-weighting temperature factor.”**
>
> We thank the reviewer for raising this concern. For the label correction threshold, we have set it to be $0.9$ in all our experiments. For the re-weighting temperature factor $\gamma$, we search within $\{3.0, 2.0, 1.5, 1.0\}$. The validation set sizes used for evaluating these hyper-parameter choices range from 536 to 5120 (cf. Table 1). We will include a discussion of the search space and search cost in the main text in the updated paper.
>
> **References**
>
> - Jiang, Y., Neyshabur, B., Mobahi, H., Krishnan, D., & Bengio, S. (2019). Fantastic generalization measures and where to find them. *arXiv preprint arXiv:1912.02178*.

---

### Decision · Program_Chairs · 2021-09-28

**Decision:**

Accept (Poster)

**Comment:**

Most reviewers and myself expressed a positive opinion on the paper following the discussion phase (which greatly helped reach a decision), and its interest to the NeurIPS community and significance of the results. The reviewers however asked for a number of clarifications and I would like to stress that the authors *must* revise significantly their initial submission according to the reviews and the discussion.

**Consistency Experiment:**

NeurIPS has a long history of experimentation. In 2014, NeurIPS ran an experiment in which 10% of submissions were reviewed by two independent committees to quantify the randomness in the review process. This year, we repeated a variant of this experiment to see how the quality of the review process has changed over time.  This paper was part of the experiment and was therefore assigned to two committees (consisting of reviewers, an Area Chair, and a Senior Area Chair) that reached independent decisions.  If both committees made the same recommendation, this recommendation was followed. If a single committee recommended acceptance, the paper was accepted (with the exception of a few cases in which the other committee identified what we considered a fatal flaw, e.g., an error in a key result).

This copy’s committee reached the following decision: **Accept (Poster)**

The other committee assigned to the paper recommended **Reject**.  You can find the other set of reviews, along with any follow up discussion with the authors here:
https://openreview.net/forum?id=j7buX9nsfis